# Clarifying the role of an unavailable distractor in human multiattribute choice

Yinan Cao[1]\*, Konstantinos Tsetsos[1,2]\*

[1]Department of Neurophysiology and Pathophysiology, University Medical Center Hamburg-Eppendorf, Hamburg, Germany; [2]School of Psychological Science, University of Bristol, Bristol, United Kingdom

**Abstract** Decisions between two economic goods can be swayed by a third *unavailable* 'decoy' alternative, which does not compete for choice, notoriously violating the principles of rational choice theory. Although decoy effects typically depend on the decoy's position in a multiattribute choice space, recent studies using risky prospects (i.e., varying in reward and probability) reported a novel 'positive' decoy effect operating on a single *value* dimension: the higher the 'expected value' (EV) of an unavailable (distractor) prospect was, the easier the discrimination between two available target prospects became, especially when their expected-value difference was small. Here, we show that this unidimensional distractor effect affords alternative interpretations: it occurred because the distractor's EV covaried positively with the subjective utility difference between the two targets. Looking beyond this covariation, we report a modest 'negative' distractor effect operating on subjective utility, as well as classic multiattribute decoy effects. A normatively meaningful model (selective integration), in which subjective utilities are shaped by *intra-attribute* information distortion, reproduces the multiattribute decoy effects, and as an epiphenomenon, the negative unidimensional distractor effect. These findings clarify the modulatory role of an unavailable distracting option, shedding fresh light on the mechanisms that govern multiattribute decisions.

**\*For correspondence:**
ycaoneuro@gmail.com (YC);
k.tsetsos@bristol.ac.uk (KT)

**Competing interest:** The authors declare that no competing interests exist.

## Editor's evaluation

This study presents an important finding on the decoy effect in multiattribute economic choices in humans. It makes a compelling case for the conclusion that the distractor effect reported in previous articles was confounded with the additive utility difference between the available alternatives. Though the contribution is somewhat narrowly focused with respect to the phenomenon that it addresses – the distractor effect in risky choice – it is important for understanding this particular phenomenon.

## Introduction

Humans strive to make good decisions but are rarely veridical judges of the world. Instead, our judgements are swayed by seemingly irrelevant factors (*Stone and Thompson, 1992*) and our preferences are improvised as we go along (*Summerfield and Tsetsos, 2015*). For instance, echoing well-documented optical illusions, we may perceive the same doughnut as larger on a tiny plate than on a bigger one (*Figure 1a*). Analogous distortions are encountered when we choose among alternatives that vary in more than one dimension or *attribute*. For instance, in the attraction effect (*Huber et al., 1982*), adding a similar but less advantageous 'decoy' option (a disk drive with a good storage capacity but at a high price) can make a target alternative (a lower-priced disk with more storage) appear more appealing than a competing alternative (an affordable disk with lower storage).

**Figure 1.** Context effects in decision-making. (**a**) Judgement about object size can be influenced by the surround. (**b**) Documented positive (***Chau et al., 2014***) or negative (***Louie et al., 2013***) distractor effects occur at the level of unidimensional value. A higher-valued distractor ($D_H$), compared with a lower-valued distractor ($D_L$), can *increase* the choice rate of A relative to B (*positive* distractor effect) but can also *decrease* the choice rate of A relative to B (*negative* distractor effect), leading to opposing distractor effects in past reports (***Chau et al., 2014***; ***Louie et al., 2013***). (**c**) During risky choices, if people derive subjective utilities of prospects by integrating reward magnitude and probability additively (additive utility or AU), and if targets $\{A_2, B_2\}$ are more frequently paired with distractor $D_H$ while targets $\{A_1, B_1\}$ with $D_L$, then a positive distractor effect could be explained by additive integration (because $\Delta AU_1 < \Delta AU_2$) rather than the distractor's expected value (EV: magnitude × probability). Curves: EV indifference lines. (**d**) Similarly, relative preference changes for target alternatives A and B can be driven by an *attraction-effect* bias (***Huber et al., 1982***; ***Dumbalska et al., 2020***) that depends on the position of the decoy D with respect to each target alternative. If $D_H$ appears more often closer to A while $D_L$ being more often closer to B, a positive distractor effect will ensue as a by-product of the attraction effect. In all illustrations, A-alternatives are always better than B-alternatives, with D-alternatives denoting the distractor (decoy) alternatives.

The attraction effect together with related contextual *preference-reversal* phenomena violate the very notion of rational-choice explanation according to which preferences should be menu-invariant (***Luce, 1977***; ***Rieskamp et al., 2006***). Instead, these phenomena suggest that preferences are menu-variant or *context-sensitive*, with the subjective utility (a notion of attractiveness) of an economic alternative being malleable to the properties of other available alternatives. Preference reversals have been typically reported in multiattribute choice settings, instigating a departure from 'value-first' utility theories (***Vlaev et al., 2011***)—in which attributes are integrated within each option independently—towards the development of novel theories of multiattribute choice (***Busemeyer et al., 2019***; ***Hunt et al., 2014***; ***Tsetsos et al., 2010***; ***Turner et al., 2018***; ***Usher and McClelland, 2004***).

In contrast to multiattribute decisions, decisions over alternatives varying along a single attribute have been typically regarded as complying with the principles of rational choice theory (***Bogacz et al., 2006***). A plausible account for this dichotomy could be that multiattribute decisions are more complex. Recently however, a new type of context effect has been reported in unidimensional multialternative decisions. In the so-called *negative distractor* effect, humans become more indifferent between two unequal alternatives of high reward value when a third distracting option has a high, as opposed to a low, reward value (***Louie et al., 2013***; ***Figure 1b***; although see a recent replication failure of this effect, ***Gluth et al., 2020a***). This negative distractor effect is theoretically important because it shows that violations of rationality may not be restricted to complex multiattribute decisions (***Carandini and***

*Heeger, 2011*; *Louie et al., 2013*). In turn, this motivates the development of a cross-domain, unified theory of decision irrationality.

Interestingly, in a series of more recent experiments involving risky prospects (alternatives offering reward outcomes with some probabilities), the opposite pattern was predominantly documented (*Chau et al., 2014*; *Chau et al., 2020*; *Figure 1b*): the decision accuracy of choosing the best of two target prospects is particularly compromised by a low expected-value (EV), 'unavailable' (distractor) prospect that should have never been in contention. This disruption effect fades away as the distractor's EV increases, leading to a *positive* distractor effect. However, *Gluth et al., 2018* casted doubts on this positive distractor effect, claiming that it arises due to statistical artefacts (*Gluth et al., 2018*). Later, in return, *Chau et al., 2020* defended the positive distractor effect by emphasising its robust presence in difficult trials only (it tends to reverse into a negative distractor effect in easy trials, i.e., distractor effects *interact* across different parts of the decision space; *Chau et al., 2020*).

Overall, the abovementioned risky-choice studies paint a rather complex empirical picture on the ways a distractor alternative impacts decision quality. Notably, in these studies, key design aspects and analyses rested on the assumption that participants' decisions were fully guided by the EV of the prospects. However, since Bernoulli's seminal work, it has been well established that EV is a theoretical construct that often fails to adequately describe human decisions under uncertainty (*Von Neumann and Morgenstern, 2007*; *Bernoulli, 1954*). Instead, human decisions under uncertainty are guided by subjective utilities that are shaped by a plethora of non-normative factors (*Tversky and Kahneman, 1992*). Such factors—including the absolute reward magnitude of a prospect (*Pratt, 1964*), the risk-iness of a prospect (*Weber et al., 2004*), or even the sum of the normalised reward and probability (*Rouault et al., 2019*; *Stewart, 2011*; *Farashahi et al., 2019*)—can perturb people's valuations of risky prospects in ways that the expected-value framework does not capture (*Peterson et al., 2021*). We reasoned that if such factors covaried with the EV of the distractor, then the distractor effects reported previously could, partially or fully, afford alternative interpretations.

It could be argued that if the experimental choice-sets are generated randomly and afresh for each participant, such covariations between the distractor's EV and other factors are likely to be averaged out. However, we note that all previous studies reporting positive and interacting distractor effects (*Chau et al., 2014*; *Chau et al., 2020*) used the exact same set of trials for all participants. This set of trials was generated pseudo-randomly, by resampling choice-sets until the EV of the distractor was sufficiently decorrelated from the EV difference (choice *difficulty*) between the two targets. On the positive side, presenting the same set of trials to all participants eliminates the impact of stimulus variability on the group-level behaviour (*Lu and Dosher, 2008*). On the negative side, using a single set of trials in conjunction with this decorrelation approach, increases the risk of introducing unintended confounding covariations in the elected set of trials, which in turn will have consistent impact on the group-level behaviour.

Here, we outline two classes of unintended covariations that could potentially explain away distractor effects in these datasets. First, the EV of the distractor could potentially covary with specific (and influential to behaviour) reward/probability regularities in the two target prospects (we call them *target-related* covariations). For instance, if people valuate prospects by simply adding up their payoff and probability information (hereafter *additive utility* or *AU*) (*Rouault et al., 2019*; *Farashahi et al., 2019*; *Bongioanni et al., 2021*), two target prospects both offering £1 with probability of 0.9 vs. 0.8, respectively (i.e., EV difference: 0.1, AU difference: 0.1), will be less easily discriminable than two other prospects both offering £0.5 with probability of 0.5 vs. 0.3, respectively (i.e., EV difference: 0.1, AU difference: 0.2; *Figure 1c*). Crucially, if the first choice-set (AU difference: 0.1) is more frequently associated with low-EV distractors than the second one (AU difference: 0.2), then a positive distractor effect could be attributable to additive integration rather than the distractor's EV (*Figure 1c*). Second, the EV of the distractor alternative could covary with its relative position in the reward-probability space (we call these *distractor-related* covariations). This would influence decision accuracy because, as outlined earlier, the relative position of a decoy alternative in the multiattribute choice space can induce strong choice biases (*Tsetsos et al., 2010*; *Turner et al., 2018*). For illustration purposes only, we assume that the distractor boosts the tendency of choosing a nearby target akin to the attraction effect (*Huber et al., 1982*; *Dumbalska et al., 2020*; *Figure 1d*). Under this assumption, if distractors with high EVs happen to appear closer to the correct target (i.e., the target with the highest EV), a positive distractor effect could be entirely attributable to the attraction effect.

The aim of this paper is to re-assess distractor effects in the relevant, previously published datasets (*Chau et al., 2014*; *Gluth et al., 2018*) while looking beyond these two classes of potentially confounding covariations (target- and distractor-related). We began with establishing that the first class of target-related covariations is indeed present in the examined datasets, with positive and interacting 'notional' distractor effects being evident even in matched *binary* trials, in which the distractor alternative was not present. Using a novel baselining approach, we asked if there are residual distractor effects when the influence of these unintended covariations is controlled for, reporting that distractor effects are eradicated. We then pinpointed these target-related confounding covariations to people's strong propensity to integrate reward and probability additively, and not multiplicatively. Moving forward, defining the key target and distractor *utility* variables, not using EV, but subjective (additive) utility, revealed a modest negative distractor effect. Moving on to the second class of distractor-related covariations, we established that choice accuracy was lawfully influenced by the position of the distractor in the multiattribute space (*Figure 1d*), consistent with a large body of empirical work in multiattribute choice (*Dumbalska et al., 2020*; *Tsetsos et al., 2010*). This 'decoy' influence was most pronounced when the distractor alternative was close to the high-EV (correct) target in the multiattribute space, yielding an attraction effect (i.e., a boost in accuracy) when the distractor was inferior, and a repulsion effect (i.e., a reduction in accuracy) when the distractor was superior (in subjective utility) to the target. Of note, this decoy influence peaking around the high-EV target, essentially re-describes a negative distractor effect without evoking ad hoc 'unidimensional distractor' mechanisms other than those that are needed to produce classic multiattribute context effects (*Tsetsos et al., 2010*; *Tsetsos et al., 2016*).

Overall, our analyses update the state-of-the-art by suggesting that, when confounding covariations are controlled for, only a modest negative distractor effect survives. Further, it is conceivable that this distractor effect mirrors asymmetric classic multiattribute context effects, which occurred robustly in the examined datasets.

## Results

We re-analysed five datasets from published studies (*Chau et al., 2014*; *Gluth et al., 2018*; $N = 144$ human participants) of a speeded multiattribute decision-making task. Participants were shown choice alternatives that varied along two distinct reward attributes: reward magnitude ($X$) and reward probability ($P$), which mapped to visual features (colourful slanted bars as options in *Figure 2a*). After learning the feature-to-attribute mapping, participants were instructed to choose the most valuable one among multiple options placed in this multiattribute space (*Figure 2a*) on each trial and to maximise the total reward across trials. In the ternary-choice condition, one option (the 'distractor', or $D$) was flagged *unavailable* for selection early in the trial. The expected value (or $EV$, i.e., $X$ multiplied by $P$) of the higher- and lower-value *available* alternatives ('targets' $H$ and $L$, respectively), and of the *unavailable* distractor were denoted by $HV$, $LV$, and $DV$, respectively.

Rational choice theory posits that an unavailable D should not in any way alter the relative choice between two available H and L targets. However, the behavioural data in this task (*Chau et al., 2014*) challenged this view. By examining the probability of H being chosen over L in ternary trials, that is, the *relative* choice accuracy, *Chau et al., 2014*; *Chau et al., 2020* reported that a *relative* distractor variable 'DV − HV' (the EV difference between the distractor and the best available target) altered relative accuracy. We note that other recent studies, using different stimuli, quantified the distractor influence by means of an *absolute* distractor variable DV (*Louie et al., 2013*; *Gluth et al., 2020a*). Wherever possible, we quantify distractor effects using both the relative (DV − HV) and absolute (DV) distractor variables.

To begin with, we note that, in addition to ternary trials in which D was present (but unavailable), participants encountered 'matched' binary trials in which only the H and L target alternatives were shown. These binary trials are ideally suited for assessing the extent to which previously reported distractor affects can be attributed to covariations between the distractor variable and target-related properties. This is because, for each ternary trial, we can derive a *respective* binary-choice baseline accuracy from binary trials that had the exact same targets but no distractor (*Figure 2—figure supplement 1c*). Given that participants never saw D in binary trials, the distractor variable is *notional* and should have no effect on the binary baseline accuracies. However, if D does paradoxically 'influence' binary accuracies, then this would signal that the distractor variable covaries with other unspecified

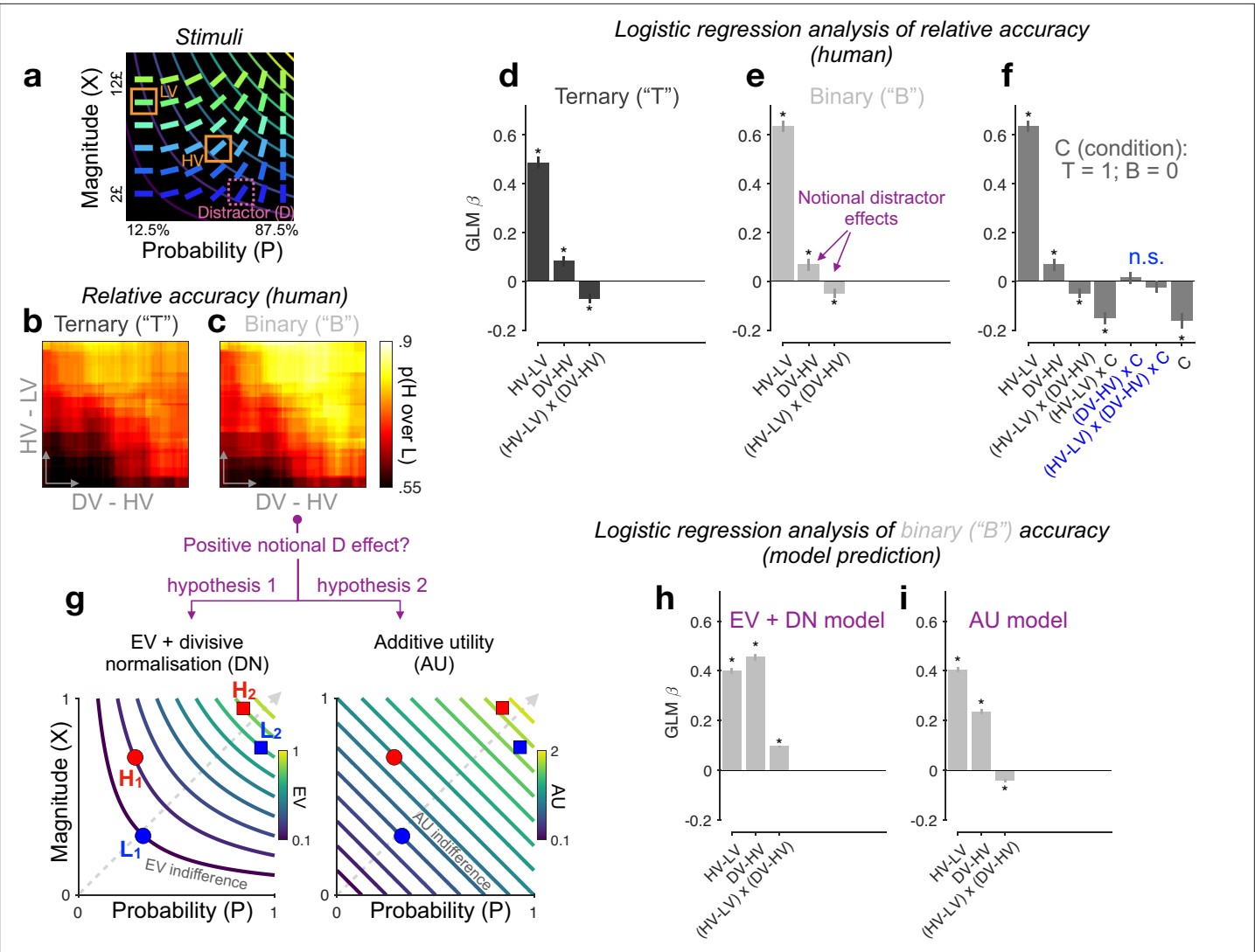

**Figure 2.** Re-assessing distractor effects beyond target-related covariations in ternary- and binary-choice trials. (**a**) Multiattribute stimuli in a previous study (*Chau et al., 2014*). Participants made a speeded choice between two available options (HV: high expected value; LV: low expected value) in the presence (ternary) or absence (binary) of a third distractor option ('*D*'). Three example stimuli are labelled for illustration purposes only. In the experiment, D was surrounded by a purple square to show that it should not be chosen. (**b, c**) Relative choice accuracy (probability of H choice among all H and L choices) in ternary trials (panel b) and in binary-choice baselines (panel c) plotted as a function of both the expected value (EV) difference between the two available options (HV − LV) (y-axis) and the EV difference between D and H (x-axis). Relative choice accuracy increases when HV − LV increases (bottom to top) as well as when DV − HV increases (left to right, i.e., *positive* D effect). (**d–f**) Predicting relative accuracy in human data using regression models (Methods). Asterisks: significant effects [p < 0.05; two-sided one-sample *t*-tests of generalised linear model (GLM) coefficients against 0] following Holm's sequential Bonferroni correction for multiple comparisons. n.s.: non-significant. (**g**) Rival hypotheses underlying the positive notional distractor effect on binary-choice accuracy. Left: EV indifference contour map; Right: additive utility (AU) indifference contour map. Utility remains constant across all points on the same indifference curve and increases across different curves in evenly spaced steps along the direction of the dashed grey line. Decision accuracy scales with Δ(utility) between H and L in binary choices. Left (hypothesis 1): Δ(EV) = (HV − LV)/(HV + LV) by virtue of divisive normalisation (DN); because $HV_2 − LV_2 = HV_1 − LV_1$, and $HV_2 + LV_2 > HV_1 + LV_1$, $Δ(EV_2)$ becomes smaller than $Δ(EV_1)$. Right (hypothesis 2): Δ(AU) = $AU_H − AU_L$; $Δ(AU_2) < Δ(AU_1)$ by virtue of additive integration. (**h, i**) Regression analysis of model-predicted accuracy in binary choice. EV + DN model corresponds to hypothesis 1 whilst AU model corresponds to hypothesis 2 in panel g. Error bars = ± standard error of the mean (SEM) (N = 144 participants).

The online version of this article includes the following figure supplement(s) for figure 2:

**Figure supplement 1.** Distinctions between our baselining approach and the method in the original study (*Chau et al., 2014*).

**Figure supplement 2.** Re-assessing distractor effects beyond target-related covariations.

**Figure supplement 3.** Re-assessing distractor effects beyond target-related covariations.

modulators of choice accuracy. We dub any effect that D has upon binary-choice accuracy as the 'notional distractor effect'. We emphasise here that a notional distractor effect is not a genuine empirical phenomenon but a tool to diagnose target-related covariations in the experimental design.

## Re-assessing distractor effects beyond target-related covariations

We used logistic regression (generalised linear model or GLM) to quantify the effect of the distractor variable on relative choice accuracy (the probability of H choice among all H and L choices). Differently from previous studies, which focused primarily on ternary-choice trials, here we analysed two different dependent variables (*Figure 2d—f*): ternary-choice relative accuracies ('T') and their *respective* baseline accuracies ('B') (see Methods and *Figure 2—figure supplement 1c*). For the ternary-choice condition, our GLM reveals a significant main effect of the relative distractor variable 'DV − HV' ($t(143) = 3.89$, p < 0.001), and a significant interaction between this distractor variable and the index of decision difficulty 'HV − LV' on relative choice accuracy, $t(143) = -3.67$, p < 0.001 (*Figure 2d*). These results agree with previous reports when analysing ternary choice alone (*Chau et al., 2014*; *Chau et al., 2020*). Turning into the matched binary baseline accuracies, the GLM coefficients bear a striking resemblance to those of the ternary-choice GLM (see *Figure 2e*; also compare *Figure 2b* to *Figure 2c* for a stark resemblance between T and B accuracy patterns), with (this time) *notional* positive and interacting distractor effects being observed. Crucially, neither the main distractor nor the (HV − LV) × (DV − HV) interaction effects are modulated by 'Condition' in a GLM combining T and B together, $|t(143)| < 0.92$, p > 0.72 (*Figure 2f*). We see similar results when the relative distractor variable DV − HV is replaced by the absolute distractor variable DV or when a covariate HV + LV is additionally included: the distractor's EV had no differential effect on relative accuracy across binary vs. ternary condition (*Figure 2—figure supplements 2 and 3*). These results equate the distractor effects in ternary trials (D present) with the notional distractor effects in binary trials (D absent), indicating that the former arose for reasons other than the properties of D. Specifically, the notional distractor effects (*Figure 2e*) in binary trials (with only H and L stimuli present) indicate that the value of D covaries with target-related properties (other than HV − LV, which was already a regressor) that modulate choice accuracy. Next, we use computational modelling to unveil these confounding target-related properties.

## Integrating reward and probability information additively in risky choice

The original study reported a surprising phenomenon (*Chau et al., 2014*): decisions seem to be particularly vulnerable to disruption by a third low value, as opposed to high value, distracting alternative, that is, a positive distractor effect. The very definition of this effect hinges upon the *a priori* assumption that decisions rely on calculating and subsequently comparing the EVs across choice options. An alternative prominent idea is that participants eschew EV calculations (*Hayden and Niv, 2021*); instead, they compute the difference between alternatives *within* each attribute and simply sum up these differences across attributes. As mentioned in the Introduction, we refer to this class of strategies that involve the additive (independent) contributions of reward and probability as the *AU* strategy. Indeed, past studies in binary risky choice have reported decision rules in humans and other animals (*Rouault et al., 2019*; *Farashahi et al., 2019*; *Bongioanni et al., 2021*) based on a weighted sum of the attribute differences, that is, a decision variable equivalent to the AU difference between alternatives, $\Delta(AU) = \lambda(HX - LX) + (1 - \lambda)(HP - LP)$, where $0 \leq \lambda \leq 1$. Although the additive combination of reward and probability may not be generalisable in all types of risky-choice tasks, it could viably govern decisions in the simple task illustrated in *Figure 2a*. Of note, we came to notice that the key distractor variable DV − HV positively covaries with the $\Delta(AU)$ between H and L across all choice-sets (e.g., Pearson's $r(148) = 0.314$, p < 0.0001, in the case of equal weighting between X and P, $\lambda = 0.5$). This unintended covariation arose in the original study possibly due to a deliberate attempt to decorrelate two EV-based quantities (DV − HV and HV − LV) in the stimuli (*Chau et al., 2014*). The correlation between DV − HV and $\Delta(AU)$ is stronger in more difficult trials ($r(71) = 0.52$, p < 0.0001, shared variance = 27%; splitting 150 trials into difficult vs. easy by the median of HV − LV; *Chau et al., 2020*) than in easier trials ($r(59) = 0.43$, p < 0.001, shared variance = 19%), mirroring both the positive (overall positive correlation) and the interacting (change of correlation strength as a function of difficulty) notional distractor effects on binary choice.

## Additive integration explains notional distractor effects in binary-choice trials

Notably, DV − HV also negatively covaries with HV + LV (Pearson's $r(148) = −0.78$, p < 0.001), which potentially leads to another explanation of why binary-choice accuracy seems lower as the matched DV − HV variable decreases. This explanation appeals to the divisive normalisation (DN) model based on EV (*Louie et al., 2013*; *Pirrone et al., 2022*). Imagine choosing between two prospects, H and L, in two different choice-sets, {$H_1$ vs. $L_1$} and {$H_2$ vs. $L_2$}, with the EV difference between H and L being the same across the two choice-sets (*Figure 2g*, left panel). DN applied to EVs ('EV + DN', hereafter) will shrink the EV difference between H and L more aggressively for set 2 than for set 1 because of the larger denominator in set 2, rendering set 2 a harder choice problem (lower accuracy). It is important to note that, in line with previous analyses (*Chau et al., 2014*; *Gluth et al., 2018*), adding the HV + LV covariate to the GLMs does not explain away the interacting notional distractor effects in binary trials (*Figure 2—figure supplement 3*). Qualitatively, the above two hypotheses ('AU' vs. 'EV + DN') both predict a positive main notional distractor effect on binary-choice accuracy, but their predictions for the (HV − LV) × (DV − HV) interaction could differ. For instance, DN might even predict a slightly positive, rather than negative, interacting notional distractor effect on binary accuracy—in this stimulus set, the divisively normalised Δ(EV) happens to be more positively correlated with DV − HV in easier trials (shared variance = 39.8%) than in harder trials (shared variance = 36.7%).

It is also important to note that AU on its own can approximate this EV sum effect by nearly tripling the utility difference between H and L in set 1 compared with that in set 2, which also renders set 2 more difficult (*Figure 2g*, right panel). These two models can thus mimic each other. To better distinguish between these candidate hypotheses, we fit a model with a softmax decision rule to each participant's binary-choice probabilities (Methods). As expected, both the EV + DN model and the AU model (with a free $\lambda$ parameter; mean $\lambda$ estimate: 0.46, SE: 0.017) predict a positive 'notional distractor' main effect on binary accuracy (*Figure 2h, i*). However, the AU model, but not the EV + DN model, reproduces a negative notional (HV − LV) × (DV − HV) interaction effect on accuracy (*Figure 2i*), mirroring the human data (*Figure 2e*). The AU model thus qualitatively provides a parsimonious account of why notional distractor effects occurred in binary trials.

Next, we used formal Bayesian model comparison to quantitatively identify the simplest possible model that can reproduce the binary-choice behaviour. For completeness, we added in the comparison a naive expected-value (EV) model, and a recently proposed dual-route model, which relies on EV and flexibly arbitrates between DN and mutual-inhibition (MI) processes (*Chau et al., 2020*). A vanilla DN model can be viewed as a nested model within the dual-route model. All models were fitted to each participant's binary choices and reaction times (RTs) by optimising a joint likelihood function (Methods). Qualitatively, *Figure 3a* shows that the naive EV model fundamentally deviates from the human choice data: the model predicts a vertical gradient of choice accuracy constrained by HV − LV. When comparing these models head-to-head using a cross-validation scheme, we find that the AU model wins over any other model (*Figure 3b*; protected exceedance probability $P_{pexc} > 0.99$; Methods). This result still holds robustly when including, in all models, subjective non-linear distortions of attributes (*Zhang and Maloney, 2012*; 'Non-linear' in *Figure 3b, c* right panels; Methods). Moreover, the RTs predicted by the DN model mismatch the human RTs (*Figure 3a*: 'EV + DN'). But this is not the sole reason why DN fails in the model comparisons. DN still fails to win over the AU model even when we consider static versions of the models fitted to the choice probabilities alone while ignoring the RTs (*Figure 3c*, 'Static models'; Methods). These systematic model comparisons thus quantitatively support the AU model as a remarkably simple account of the 'notional distractor' effects in binary trials (*Figure 2i*). Additional model comparisons corroborate that the AU model is favoured over specific and popular instantiations of the non-linear multiplicative model (i.e., expected utility [*Von Neumann and Morgenstern, 2007*] and prospect theory [*Kahneman and Tversky, 1979*]; see *Figure 3—figure supplement 1*).

The success of the AU model suggests that the reward attributes were not multiplicatively conglomerated into a single EV dimension. To better illustrate the differences between AU- and EV-based models, we plotted model predictions separately for different condition categories in which H dominates L in one attribute only (single dom.) or in both attributes simultaneously (double dom.; only 19% of all trials) (*Figure 3d—f*). The EV model, being only sensitive to the difference in the product of probability and reward, fails to capture any performance variations across these distinct categories

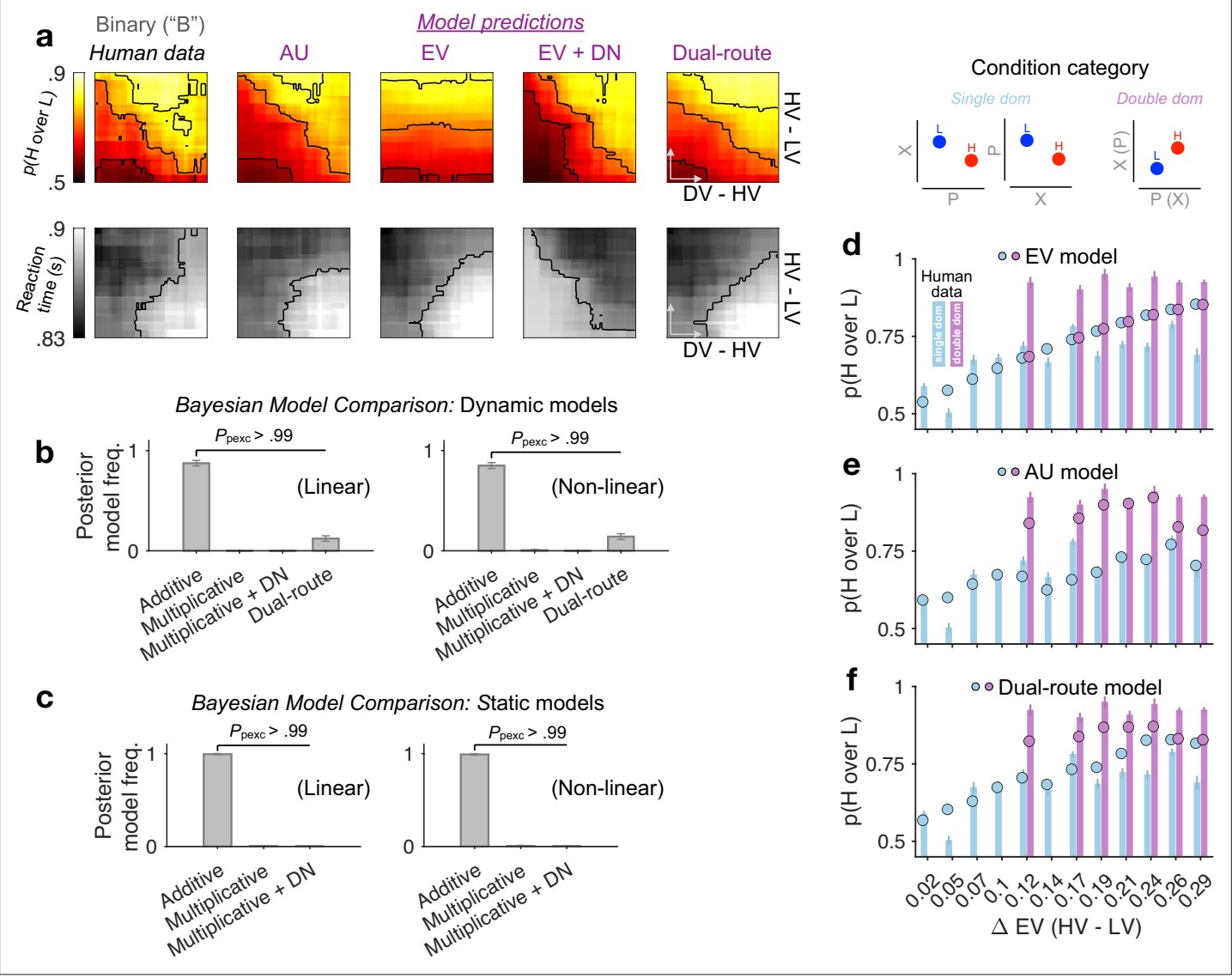

**Figure 3.** Modelling binary-choice behaviour. (**a**) Human vs. model-predicted choice accuracy and reaction time (RT) patterns. (**b, c**) Bayesian model comparison based on cross-validated log-likelihood. Linear and non-linear: different psychometric transduction functions for reward magnitude and probability (see Methods). Linear multiplicative: EV; linear additive: AU. Dynamic model: optimisation based on a joint likelihood of choice probability and RT; Static model: optimisation based on binomial likelihood of choice probability. (**d–f**) Model predictions (circles) plotted against human data (bars) for different condition categories [H dominates L on one attribute only, that is, *single dom.* (81% of all trials), or on both attributes, that is, *double dom.* (only 19%)] at each level of HV – LV. Error bars = ± standard error of the mean (SEM) (N = 144 participants).

The online version of this article includes the following figure supplement(s) for figure 3:

**Figure supplement 1.** Additional multiplicative models (expected utility and prospect theory).

(because these categories are orthogonal to the EV difference; *Figure 3d*). By contrast, the AU model can capture these across-category performance changes (*Figure 3e*). Qualitatively, the dual-route model seems to be able to capture some patterns of the human data on the binned maps (*Figure 3a*: 'Dual-route'), but its deviation from human behaviour becomes more salient when we inspect the model predictions at each fixed level of EV difference (*Figure 3f*). Taken together, we show that the AU model reproduces the patterns of both accuracy and RT in binary-choice trials decisively better than the models relying on EV or divisively normalised EV (also see *Supplementary file 1*, *Supplementary file 2* for model parameter estimates).

## A modest negative distractor effect operating on AU

Because 'additive integration' is a decisively better model of binary-choice behaviour than multiplicative models are (e.g., EV, EV + DN, expected utility, and prospect theory), here we re-assessed distractor effects after defining the key analysis variables using the subjective utilities afforded by the AU model. To better define these variables, we fit the dynamic AU model (with free $\lambda$ weight; Methods: *Context-independent model*) to binary and ternary choices, separately, and constructed subjective utility as AU = $\lambda(X) + (1 - \lambda)(P)$. We then analysed relative choice accuracy using a GLM with AU-based regressors: $AU_H - AU_L$ (AU difference between targets), $AU_H + AU_L$ (AU sum), distractor variable (absolute: $AU_D$, or relative: $AU_D - AU_H$), distractor variable × ($AU_H - AU_L$) interaction, and the interaction between each of these effects and 'C' (Condition: T vs. B). We focus on the key regressors that assess the distractor effects in ternary choices over and above the matched baseline performance (captured by the '× C' interaction): the main distractor effect, 'distractor variable × C', and the interacting distractor effect dependent on choice difficulty, 'distractor variable × ($AU_H - AU_L$) × C'. First, we verified that including the AU-sum regressors decisively improved the GLM's goodness-of-it, $\chi^2$ (2) > 1.005 × 10³, p < 0.0001; and the GLM with AU regressors had better goodness-of-it than the GLM with EV regressors, $\chi^2$ (0) > 1.03 × 10⁴, p < 0.0001 (for either absolute or relative distractor variable). We found a significant *negative* main distractor effect, $AU_D$ × C, on the choice accuracy, t(143) = −2.27, p = 0.024 (uncorrected); however, this main effect was marginally significant when using the relative distractor variable $AU_D - AU_H$, t(143) = −1.86, p = 0.065 (uncorrected). The interacting distractor effect was not significant t(143) < −0.54, p > 0.1 (see *Supplementary file 3* for a full report). We here reiterate that, by contrast, neither the main nor the interacting distractor effects were significant in the GLM with EV-based regressors (p > 0.3, uncorrected, *Figure 2f* and *Figure 2—figure supplements 2 and 3*). These results based on subjective AU fit more closely with recent reports showing a *negative* main distractor effect (*Louie et al., 2013*).

## Distractor-related covariations and multiattribute context effects

Having revisited distractor effect while controlling for target-related confounding covariations, we now move into examining distractor-related covariations, that is, whether multiattribute context effects were present in these datasets, and if so, whether they were related to the modest AU-based negative distractor effect we reported above.

Classic multiattribute context effects typically occur when a decoy alternative is added to a choice-set consisting of the two non-decoy alternatives (*Figure 4a*). Two classic context effects are particularly relevant in this task because 84% of the ternary trials satisfy specific H, L, and D geometric (i.e., in the two-dimensional choice space) configurations that could give rise to these context effects (*Figure 4a*). First, an attraction effect (*Huber et al., 1982*), whereby the probability of choosing a target over a competitor increases due to the presence of an 'inferior' *dominated* D near the target (also see *Figure 1d* for an illustrative example); and second, a repulsion effect (*Dumbalska et al., 2020*), whereby this pattern is reversed (and the probability of choosing the target is reduced) in the presence of a nearby 'superior' *dominating* D. We note in passing that other popular context effects, such as the similarity and the compromise effects, could not be examined here due to the absence or the very low frequency of occurrence of appropriate choice-sets (i.e., in these effects the decoys are required to be roughly iso-preferable to the target or the competitor alternatives, *Spektor et al., 2021*).

Before characterising decoy effects, we considered the possibility that the change of relative accuracy between binary (B) and ternary (T) choice reflects a condition-unspecific change in behaviour, that is, an accuracy reduction potentially induced by higher cognitive demands in the ternary trials. We estimated this 'generic' bias by a permutation procedure whereby we randomly shuffled the true mappings between ternary conditions and their matched binary conditions and obtained an average T − B accuracy contrast across random permutations (*Figure 4c*; see Methods). After this bias was corrected for each participant, the relative accuracy change from B to T was compared against zero (i.e., zero represents a null hypothesis in which T and B had no difference over and above the condition-unspecific bias). The bias-corrected T − B relative accuracy change shows a strong attraction effect: The relative choice accuracy dynamically decreases and increases as an inferior D gets closer to L (*n* = 32 relevant trials in this category) and H (*n* = 33), respectively, |t(143)| > 4.88, p < 0.0001 (*Figure 4d*). Meanwhile, a significant repulsion effect occurs when a superior D dominates H or L in the

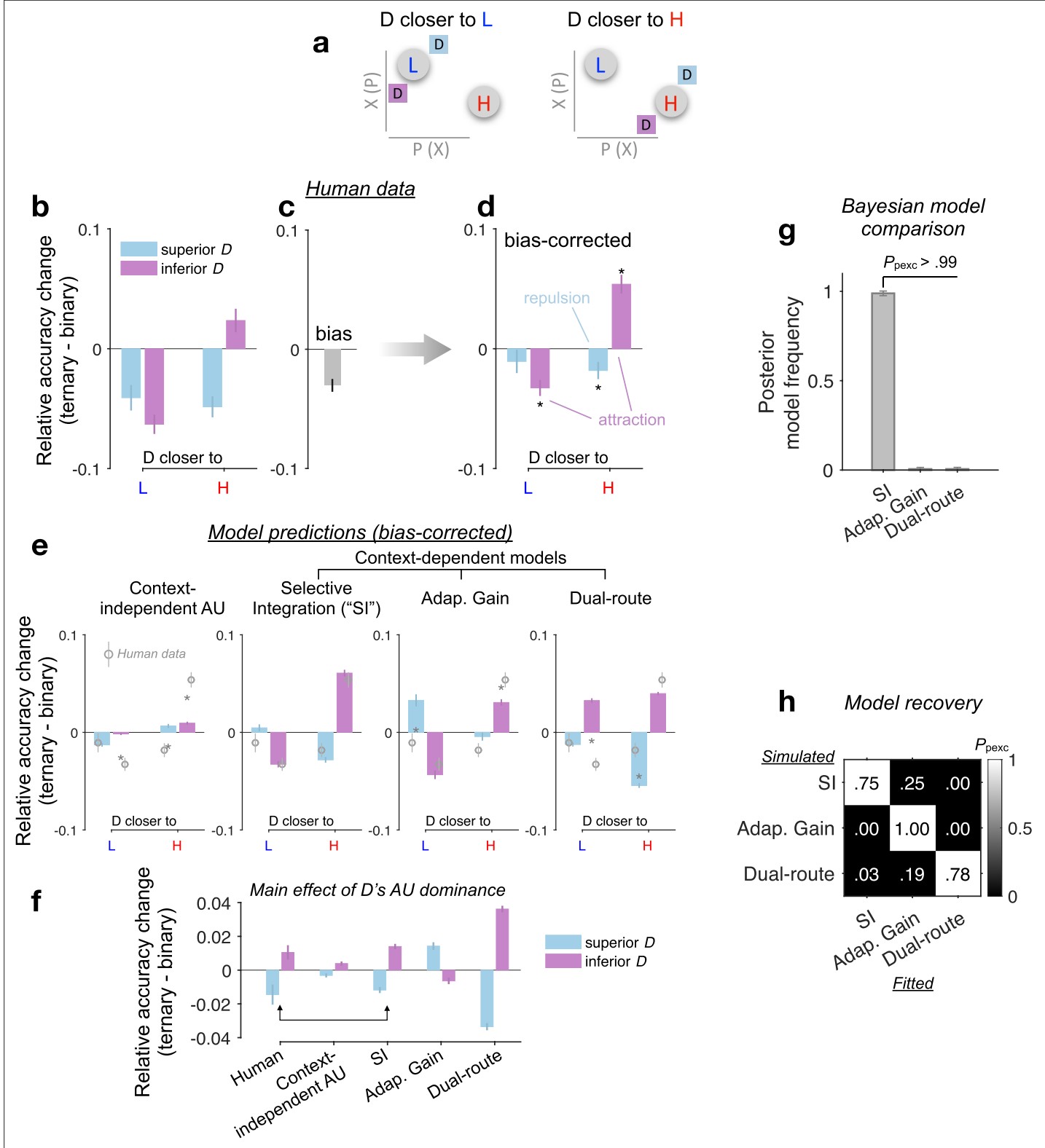

**Figure 4.** Multiattribute context effects. (**a**) The attraction effect (***Huber et al., 1982***) can happen when D is close to and dominated by H or L. The repulsion effect (***Dumbalska et al., 2020***) can happen when D is close to but dominates H or L. (**b**) T − B $p$(H over L) plotted as a function of D's relative Euclidean distance to H vs. L in the attribute space and D's additive utility (AU) dominance—'superior' ('inferior') D lies on a higher (lower) AU indifference line than both H and L. Error bars: ± standard error of the mean (SEM) across participants ($N$ = 144). (**c**) Condition-unspecific bias of $p$(H over L) in ternary conditions relative to binary conditions. (**d**) Bias-corrected T − B $p$(H over L). Black asterisks: significant context effects (Bonferroni–Holm

*Figure 4 continued on next page*

*Figure 4 continued*

corrected p < 0.05; two-tailed one-sample *t*-tests against 0). (**e**) Bias-corrected T − B *p*(H over L) predicted by different models. Grey asterisks: significant difference between model predictions and human data (uncorrected p < 0.05; two-tailed paired-sample *t*-tests). (**f**) Main effect of D's AU dominance (superior vs. inferior) on T − B relative accuracy change in human data and model predictions. (**g**) Bayesian model comparison of context-dependent models (also see *Figure 4—figure supplement 1*). (**h**) Model recovery (Methods). Text in each cell: posterior model frequency; Heatmap: protected exceedance probability ($P_{pexc}$) that a fitted model (each column) has the highest log-likelihood given input ternary-trial data (choice probabilities and RTs) simulated by a certain model (each row).

The online version of this article includes the following figure supplement(s) for figure 4:

**Figure supplement 1.** Additional modelling results.

attribute space, $t(143) = -2.53$, p = 0.025, particularly when D is closer to H (*n* = 41 relevant trials in this category), but not when D is closer L (fewer trials *n* = 20 in this category; $t(143) = -1.09$, p = 0.28, Bonferroni–Holm corrected). A repeated-measures analysis of variance (ANOVA) reveals a significant interaction between the similarity (D closer to H vs. D closer to L) and the dominance of D (inferior vs. superior), $F(1, 143) = 30.6$, p < 0.0001, partial $\eta^2 = 0.18$, on the T − B relative accuracy change.

Of note, in line with the AU-based negative distractor effect we reported in the previous section, the current analysis on multiattribute decoy effects also reveals a significant main effect of D's AU dominance on relative accuracy change (repeated-measures ANOVA on T − B relative accuracy change in all trials), $F(1, 143) = 6.51$, p = 0.012, partial $\eta^2 = 0.044$ (*Figure 4f*: 'Human'). This overall dominance effect seems to be induced by a noticeable asymmetry in the observed decoy effects (*Figure 4d*, net of inferior-purple vs. superior-blue bars): targeting H seems more effective than targeting L, and also, the attraction effect is stronger than the repulsion effect (hence, the multiattribute effects do not cancel out but lead to a net accuracy boost for inferior D's). Crucially, across participants there is a significant positive correlation between the negative main distractor effect 'AU$_D$ × C' (GLM in the previous section) and the negative main effect of D's AU dominance here (Spearman's $\rho = 0.28$, p = 0.00074). This indicates that if a participant had a stronger *asymmetry* in the decoy effects (D's AU dominance) then they would also have a stronger AU-based negative distractor effect. We further interpret this correspondence using computational modelling in the next section.

## Selective gating explains multiattribute context effects and adapts to decision noise

Although the AU model successfully described behavioural responses in binary trials (*Figure 3*), it is essentially a context-independent model complying with the *independence from irrelevant alternatives* (IIA) principle. That is, according to the AU model, the characteristics of D in ternary trials should not lead to any preference change between the two available targets H and L. In this section, we resort to computational modelling to understand why multiattribute context effects occur in ternary trials. First, as shown in *Figure 4e* leftmost panel, the T − B effects in human data could not be explained by a context-independent AU model ('blind' to D, i.e., only taking H and L as inputs) with freely varying parameters across B and T conditions (the free parameters could capture any global changes in behaviour due to, for example, different noise levels and/or attribute weighting, potentially induced by higher cognitive demands in the ternary trials; Methods).

Next, we considered two models that can predict context effects in multiattribute choice: selective integration (SI) and adaptive gain (*Dumbalska et al., 2020*; *Li et al., 2018*; Methods). In the SI model, the gain of processing on each attribute value relies on its rank within that attribute, with a selective gating parameter *w* controlling the suppression strength for the low-rank attribute value. In the adaptive gain model, each alternative is evaluated relative to the context statistics via a non-linear function. We compared these models to the context-dependent dual-route model of Chau et al., which had not been previously examined in relation to classic context effects (*Chau et al., 2020*). As shown in *Figure 4e*, the SI model can reproduce all types of attraction and repulsion effects in the data, but a dual-route model cannot. A systematic Bayesian random-effects model comparison shows that SI prevails over all other models ($P_{pexc} > 0.99$; *Figure 4g*), including the dual-route, DN, or any context-independent model with either EV or AU as the utility function (*Figure 4—figure supplement 1a*). The adaptive gain model captures some key context effects, although not as closely as the SI model does (fixed-effects: SI vs. adaptive gain, cross-validated log-likelihood ratio = 219.8, ΔBIC = −1194, pooled over participants; random-effects: SI's $P_{pexc} > 0.99$; *Figure 4e*), and explains the human data decisively

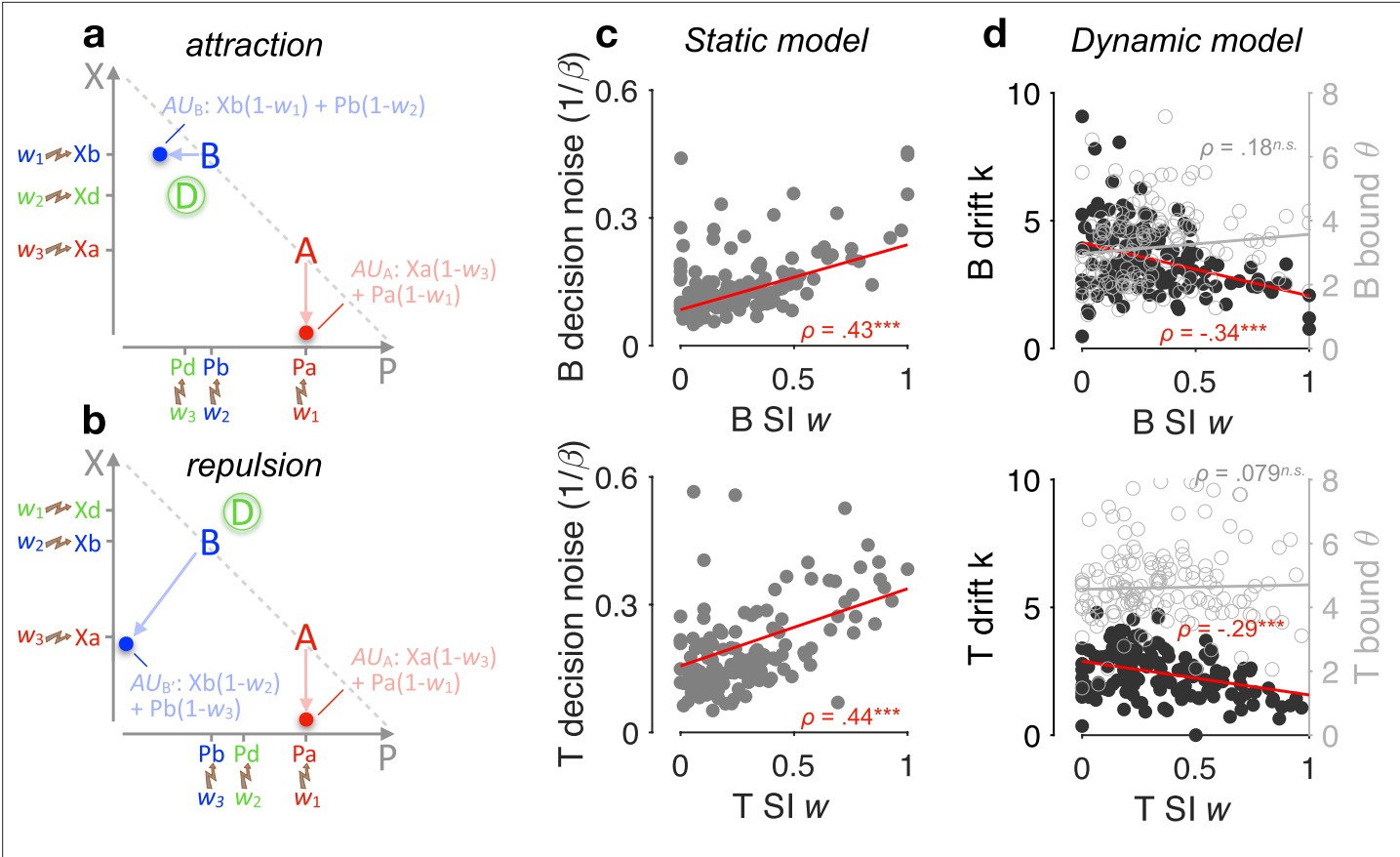

**Figure 5.** Selective gating, context effects, and decision noise. (**a, b**) By virtue of selective integration (SI), alternatives A, B, and D are ranked within each attribute dimension; SI weights are then assigned to the sorted alternatives accordingly (for illustration, $w_1 = 0$, $w_2 = 0.5$, and $w_3 = 0.9$). Attribute values are selectively suppressed in proportion to the SI weights, that is, lower ranked values are suppressed more strongly. As a result of SI (assuming any $w_1 < w_2 \le w_3$), the additive utility (AU) of B is higher in the case of panel a than the AU of B' in the case of panel b due to the distinct locations of decoy D. (**c**) Relationship between SI $w$ and decision noise. Red lines: robust linear regression fits (using Matlab's robustfit). $\beta$: softmax inverse temperature. Top: binary trials ('B'); bottom: ternary trials ('T') (ternary SI $w$: mean across $w_2$ and $w_3$); SI $w = 0$: lossless integration, that is, attributes maintain their original values; SI $w > 0$: selective integration; SI $w = 1$: losing attribute value shrinks to 0. $\rho$: Spearman's rank correlation coefficient. ***p < 0.0001. (**d**) Relationship between SI $w$ and drift sensitivity $k$ (filled black) or decision bound (open grey). Each point represents a participant ($N = 144$ in total).

The online version of this article includes the following figure supplement(s) for figure 5:

**Figure supplement 1.** Parameter recovery.

better than the dual-route model (adaptive gain vs. dual-route: cross-validated log-likelihood ratio = 48.58, $\Delta$BIC = −806, pooled over participants; adaptive gain's $P_{pexc} = 0.78$; see ***Supplementary file 4*** for model parameters). A model recovery procedure (Methods) also confirmed that our Bayesian model comparison was capable of properly discriminating among these context-dependent models (SI, adaptive gain, and dual-route; ***Figure 4h***).

Crucially, the SI model could lawfully reproduce the overall negative D's AU dominance effect without relying on any additional mechanism such as DN, $F(1, 143) = 78.4$, $p < 0.00001$, partial $\eta^2$ = 0.35, *post hoc* t-test: $t(143) = −8.85$, $p < 0.0001$ (***Figure 4f***). By contrast, the models that predict 'by-design' unidimensional distractor effect (dual-route and DN) could not describe the multiattribute decoy effects (***Figure 4e, f***; also see ***Figure 4—figure supplement 1b, c***). This suggests that the negative D dominance effect is not the by-product of a unidimensional distractor mechanism but instead falls out naturally from the presence of asymmetrical decoy effects (i.e., stronger attraction effect, when H is targeted). The correlation between the D dominance and AU distractor effects discussed above, corroborates the possibility that, to some extent, the negative distractor effect is driven by asymmetrical decoy effects (***Figure 4f***: SI model mimics the negative D dominance effect).

How does the winning SI model capture the attraction and repulsion effects? The way the model works in these two effects is shown in a schematic form in *Figure 5a, b*. Because SI selectively allocates reduced gain to 'locally losing' attribute values, it exerts a stronger gating on the utility of option B when the decoy D dominates B (*Figure 5b*) than when D is dominated by B (*Figure 5a*). Thus, the relative preference between A and B can be dynamically reversed according to the dominance of D, giving rise to the opposing context effects of attraction and repulsion.

Lastly, SI has been shown to maximise decision accuracy and economic outcomes in the face of decision noise that arises when multiple attributes or pieces of information need to be combined towards a decision (*Tsetsos et al., 2016*; *Wyart and Koechlin, 2016*). Here, we examined whether SI confers robustness on decisions by adapting to the individual decision noise levels. And if so, it could provide a normative justification of these decoy biases as emergent properties of a decision policy that maximises robustness against decision noise. After fitting a static version of the SI model to choice probabilities (Methods), we found a positive correlation across participants between the SI gating $w$ and the temperature of a softmax choice function (Spearman's $\rho(142) > 0.43$, p < 0.0001, for both binary and ternary choices; *Figure 5c*). Additional parameter-recovery simulations corroborated that these findings are not due to any trade-off between model parameters (*Figure 5—figure supplement 1*). After fitting the dynamic SI model to choices and RTs together, we found that the SI $w$ negatively correlates with the utility drift sensitivity (how effective a participant is in using the information about utility; Methods), but not the height of the decision bound (*Figure 5d*). These findings suggest that SI has an adaptive role, as it selectively exaggerates the attribute difference between winning and losing alternatives by just the right amount in proportion to decision noise.

## Discussion

Context effects violate the axioms of rational choice theory and can shape our understanding of the computational mechanisms that underlie decision-making. Recent reports suggested that such effects do not occur only in multiattribute space, but also at the level of a single value dimension (*Chau et al., 2014*; *Louie et al., 2013*; *Chau et al., 2020*; *Webb et al., 2020*). However, these reports diverge in their conclusions (see also *Gluth et al., 2020a*; *Gluth et al., 2018*; *Gluth et al., 2020b*), with some studies showing a negative distractor effect (*Louie et al., 2013*) while others, using risky-choice tasks, showing positive and interacting distractor effects (*Chau et al., 2014*; *Chau et al., 2020*). Here, we focused on these risky-choice datasets and examined whether the positive and interacting distractor effects could afford alternative interpretations. We found that the previously reported positive and interacting effects can be explained away by confounding covariations among distractor variables and subjective utility, which in these datasets was shaped by additive and not multiplicative reward/ probability integration. When we redefined distractor effects using the descriptively more adequate 'additive utility', we instead reported a modest negative distractor effect. Asides from this modest negative distractor effect, classic context (decoy) effects—determined by the multiattribute similarity and dominance relationship between the distractor and the available alternatives—occurred robustly. Of note, the multiattribute context effects occurring here were technically 'phantom' effects (*Pettibone and Wedell, 2007*) due to the fact that the distractor could not be chosen; and contrary to most previous demonstrations, where the target and competitor alternatives are almost equally valuable, the effects we reported here arise even under a noticeable value difference between the target and the competitor (*Farmer et al., 2017*). These context effects were asymmetric in magnitude, yielding an average accuracy increase for inferior distractors and an average accuracy decrease for superior ones, mirroring thus the negative distractor effect.

These results clarify the role that an unavailable distractor alternative plays in decisions between two uncertain prospects. Notably, the negative distractor effect we reported is of rather small size. A plausible explanation of this negative distractor effect is that it could be driven, to some extent, by asymmetric multiattribute context effects. We made this conjecture for two reasons. First, from an empirical perspective, the 'dominance' effect arising from multiattribute decoy effects correlates with the negative distractor effect. Second, from a theoretical perspective, model comparison favours by large a model that can only produce multiattribute decoy effects and has no dedicated mechanism to generate unidimensional distractor effects. This alludes to a parsimonious account that precludes genuine unidimensional distractor effects. However, we interpret the link between the distractor and decoy effects with caution given that there are differences in the way the two effects were quantified.

In the distractor effect, the key variable is defined continuously (i.e., the distractor's utility) while in decoy effects, 'dominance' is defined categorically (superior or inferior to the target). Similarly, a less parsimonious hybrid model in which both unidimensional distractor mechanism (i.e., DN) and multi-attribute context sensitivities (i.e., SI) coexist has not been ruled out. Definitive conclusions on the link between distractor and decoy effects await new experimental designs in which the two types of effects can be quantified independently.

In light of the stronger robust presence of decoy effects in these datasets, the hypothesis that multiattribute complexity is a key moderator of context dependencies, remains viable (*Hunt et al., 2014*; *Juechems and Summerfield, 2019*; *Fellows, 2006*). Encoding absolute information along several attributes is inefficient (*Summerfield and Tsetsos, 2015*) and suffers from the *curse of dimensionality* (*Sutton, 1988*; *Bellman, 1957*). A plausible solution to circumvent these issues is to discard information that is less important (*Niv, 2019*). In a similar spirit, our SI model preferentially focuses on larger values at the expense of smaller values, and because of this policy, context effects ensue. This selective policy potentially maximises robustness to decision noise (*Tsetsos et al., 2016*; *Luyckx et al., 2020*), providing thus a normative justification of contextual preference reversal. And although the SI model had thus far been applied to dynamic (sequential) value accumulation tasks (*Usher et al., 2019*), we here show that it can also account for behaviour in a static multiattribute task. Nevertheless, we do not claim that SI, despite its simplicity and computational tractability, is the only computational model that could explain the multiattribute context effects in this task (*Noguchi and Stewart, 2018*; *Trueblood et al., 2014*; *Bhatia, 2013*; *Roe et al., 2001*).

Notably, and contrary to the normative rule, participants in the datasets we analysed integrated probability and payoff information additively and not multiplicatively. This additive integration, or more generally, the processing of information in a 'within dimensions' fashion, is probably what spurs contextual preference reversal in risky choice. However, unlike what we observed here, a large and influential body of research in risky decision-making supports the idea that non-linear transformations of probabilities and rewards are multiplicatively combined (e.g., prospect theory, *Kahneman and Tversky, 1979*). Additive integration in risky choice has limited applicability as opposed to the more widely applied multiplicative-integration framework. Unlike multiplicative models, the additive-integration framework enforces independent contributions of reward and probability (*Koechlin, 2020*) and cannot readily accommodate rich attitudes towards risk (*Tversky and Wakker, 1995*) or other complex phenomena in decisions over multibranch gambles (*Birnbaum, 2006*; *Birnbaum, 2008*). At an even more fundamental level, additive integration requires *ad hoc* assumptions (on the way outcomes and probabilities are represented) to capture the decreasing desirability of negative outcomes with increasing probability, or risk-attitudes over mixed gambles that offer both positive and negative outcomes (*Tom et al., 2007*). Although these limitations suggest that additive integration is not readily generalisable to all types of risky choice, this does not undermine its descriptive superiority in simpler tasks like the one we focused on here. One overarching conjecture is that the stimulus employed in different experiments determines the way participants integrate reward and probability, with multiplicative integration observed in numerically explicit gambles and additive integration when rewards and probabilities are inferred from abstract non-numerical (perceptual) dimensions (*Rouault et al., 2019*; *Stewart, 2011*; *Farashahi et al., 2019*; *Bongioanni et al., 2021*; *Koechlin, 2020*; *Massi et al., 2018*; *Donahue and Lee, 2015*). This may further suggest that risk preference is a rather elusive and unstable construct (*Chater et al., 2011*).

Additive and multiplicative integration strategies are not necessarily mutually exclusive, and they could instead co-influence risky decisions (*Farashahi et al., 2019*; *Bongioanni et al., 2021*). More generally, neural correlates of both additive and multiplicative decision variables have been reported within the same task and even within overlapping brain regions. Neural signals in a network of cortical and subcortical brain regions have been shown to code for value/utility-related metrics of diverse assets (*Tom et al., 2007*; *De Martino et al., 2006*; *Platt and Glimcher, 1999*; *Kennerley et al., 2009*). Within this network, the dorsal medial frontal cortex (dmPFC in humans [*Hunt et al., 2014*] and an anterior/dorsal frontal site located between the rostral and anterior cingulate sulci in area 32 of non-human primates [*Bongioanni et al., 2021*]) seems to be particularly relevant for the multiplicative integration of reward and probability in risky decisions. Strikingly, a focal ultrasonic disruption of this anterior/dorsal frontal region (but not the posterior/ventral medial frontal cortex [vmPFC]) caused a shift from a multiplicative to an additive strategy in the macaques (*Bongioanni et al., 2021*),

imputing a direct causal link between EV computation and the dorsal medial frontal lobe. By contrast, the vmPFC seems to be involved in representing individual sources of reward information such as the latent reward probabilities or states (*Rouault et al., 2019*; *Koechlin, 2014*) and the prospective appetitive values of risky gambles (*Rouault et al., 2019*), and is causally linked to the within-attribute comparison across alternatives (*Fellows, 2006*), providing key process components for the additive strategy. Interestingly, however, the human dmPFC, but not vmPFC, has been shown to guide risky choice by combining *additively* bandits' reward probabilities with their normalised utilities (*Rouault et al., 2019*). Explicit coding of multiplicative value has also been found in the orbitofrontal cortex (OFC) and ventral striatum (*Yamada et al., 2021*), while reward attribute representations and comparisons could also engage the OFC (*Suzuki et al., 2017*) or the parietal cortex (*Hunt et al., 2014*) depending on the task and the relevant attributes. This rather complicated picture of the neural coding of multiplicative vs. additive integration suggests that, even though behaviour may be by and large best described by a particular decision strategy (*Cao et al., 2019*), the brains could afford simultaneous representations of multiple competing strategies (*Cao et al., 2019*; *Williams et al., 2021*; *Fusi et al., 2016*).

To conclude, harnessing the power of open science, we scrutinised large datasets from published studies (over 140 human participants) and re-assessed the role of the distractor alternative in decisions under uncertainty. More broadly, we adopted an integrative approach attempting to characterise the interplay between two types of context effects (unidimensional and multiattribute), which had thus far been examined in isolation. Our findings contribute to a large body of research (*Hayden and Niv, 2021*; *Noguchi and Stewart, 2018*; *Roesch et al., 2006*; *Soltani et al., 2012*) that challenges the standard tenet of neuroeconomics that value representation is menu-invariant (*Kable and Glimcher, 2009*; *Levy and Glimcher, 2012*; *Padoa-Schioppa and Assad, 2008*).

## Methods
### Multiattribute decision-making task and datasets
In the multiattribute decision-making task of *Chau et al., 2014*, participants made a speeded choice within 1.5 s between two available options (HV: high value; LV: low value) in the presence ('ternary') or absence ('binary') of a third unavailable distractor option (D). On each trial, the two or three options appeared in different quadrants of a computer screen. Each option was a small rectangle and its colour/orientation represented some level of reward magnitude/probability (the feature-to-attribute mapping was counterbalanced across participants). There were equal number of ternary ($n$ = 150) and binary trials ($n$ = 150). However, the ternary and binary trials did not have 'one-to-one' mapping, that is, some ternary trials had only 1 matched binary trial with the same H and L options while others had several (also see *Figure 2—figure supplement 1*). The available targets and the unavailable distractor were flagged by orange and pink surround boxes, respectively, 0.1-s poststimulus onset. Participants received feedback about whether the chosen option yielded a reward at the end of each trial. We re-analysed five published datasets of this speeded-decision task ($N$ = 144 human participants in total). These five datasets all include ternary and binary trials: the original fMRI dataset ($N$ = 21, *Chau et al., 2014*), a *direct* replication study (using exact trial sequence provided by Chau et al.) performed by Gluth et al. (Gluth Experiment 4: $N$ = 44), and additional replication experiments performed by Gluth et al. (Gluth Experiments 1, 2, and 3; $N$ = 79 in total, *Gluth et al., 2018*). Of note, a new dataset ('Hong Kong'; $N$ = 40) recently acquired by *Chau et al., 2020* did not contain binary trials and thus could not be included in our analyses that compared ternary and binary choices head-to-head.

### Computational modelling of behaviour
General architecture of dynamic models

We used the framework of feedforward inhibition (FFI) model to jointly fit choice probability and RT. The FFI model conceptually 'interpolates' between a race model and a drift-diffusion model (*Mazurek et al., 2003*). Each accumulator is excited by the evidence for one alternative and inhibited by the evidence for the other alternatives via feedforward inhibition. Evidence units are ideal integrators with no leak, and they receive crossed inhibition from different alternatives. The activation of each integrator is $x_i$ (for the $i$ th alternative), which takes $I_i$ as evidence input:

$$dx_i = \left(kI_i + I_0\right) dt + \xi\sqrt{dt} \tag{1}$$

where $k$ is a sensitivity parameter (drift rate coefficient, i.e., how effective the nervous system is in using utility information), and $I_0$ a baseline drift, independent of $I_i$. $\xi$ denotes Gaussian noise of a zero mean and a variance of $\sigma^2$. The noise term is in proportion to the square-root of $dt$ because the noise variance itself is a linear function of $dt$. Evidence $I_i$ is computed as the utility of alternative $i$ inhibited by the average utility across all other alternatives in a choice-set (assuming $n$ alternatives in total including $i$):

$$I_i = U_i - c\left(\frac{\sum_{j\neq i} U_j}{n-1}\right) \tag{2}$$

where c ($0 \leq c \leq 1$) is the FFI strength. When c = 0, the model becomes a classic race model; when c = 1, it corresponds to a drift-diffusion model. The drift rate $\mu$ of FFI diffusion thus can be rewritten as:

$$\mu_i = kU_i - ck\left(\frac{\sum_{j\neq i} U_j}{n-1}\right) + I_0 \tag{3}$$

The utility function $U_i$ can take different forms of interest, such as EV, AU, or being adaptive to contexts (SI and adaptive gain models, see next).

We next derive the analytical solutions for the FFI dynamic system (**Equation 1**). First, the distribution of first-passage times $t$ (i.e., decision-bound hitting times) is known to follow an inverse Gaussian distribution (**Usher et al., 2002**):

$$p\left(t; \theta, x_0, \mu\right) = \frac{\theta - x_0}{\sqrt{2\pi\sigma^2 t^3}} e^{\frac{-(\theta - x_0 - \mu t)^2}{2\sigma^2 t}} \tag{4}$$

In which, $x_0$ is a starting point of the integrator, and $\theta$, the height of decision bound. Then, we can derive the probability of observing a response time $RT$ on a given trial as:

$$p\left(t = T\right) = \frac{\theta - x_0}{\sqrt{2\pi\sigma^2 T^3}} e^{\frac{-(\theta - x_0 - \mu T)^2}{2\sigma^2 T}} \tag{5}$$

where $T = RT - t_{nd}$, and $t_{nd}$ is a non-decision response delay. Now, we can obtain the probability of observing a choice $i$ but not choice $j$ at time $T$ as:

$$p\left(\text{choice} = i, t = T\right) = p\left(t_i = T\right) p\left(t_j > T\right) = p\left(t_i = T\right)\left[1 - p\left(t_j < T\right)\right] \tag{6}$$

That is, we can multiply the PDF for response $i$ (e.g., H choice) with the cumulative probability that the alternative response $j$ (e.g., L choice) hits the decision bound if and only if at a time later than $T$. Given that a CDF of an inverse Gaussian is expressible in terms of the CDF of the standard Gaussian $\Phi(\cdot)$, that is,

$$p\left(t_j < T\right) = \Phi\left(\frac{\mu_j T - \theta + x_0}{\sigma\sqrt{T}}\right) + e^{\frac{2(\theta - x_0)\mu_j}{\sigma^2}} \Phi\left(\frac{-\mu_j T - \theta + x_0}{\sigma\sqrt{T}}\right) \tag{7}$$

and by substituting **Equation 7** into **Equation 6**, we obtain the likelihood function of observing choice $i$ at time $T$ as:

$$L\left(i, T\right) = \left[1 - \Phi\left(\frac{\mu_j T - \theta}{\sigma\sqrt{T}}\right) + e^{\frac{2\theta\mu_j}{\sigma^2}} \Phi\left(\frac{-\mu_j T - \theta}{\sigma\sqrt{T}}\right)\right] \frac{\theta}{\sigma\sqrt{2\pi T^3}} e^{\frac{-(\theta - \mu_i T)^2}{2\sigma^2 T}} \tag{8}$$

Further, we assume no bias in the starting point of accumulation, that is, $x_0 = 0$, and a unit variance of the noise term, that is, $\sigma = 1$. This is equivalent to measuring the diffusion properties such as sensitivity $k$ and bound $\theta$ in units of $\sigma$, a convention in standard diffusion models that prevents parameter trade-offs via a re-scaling of the diffusion space (**Drugowitsch et al., 2014**; **Palmer et al., 2005**). When fitting the models to data, we maximised the sum of Log($L$) ($LL$, hereafter) across trials given empirical trial-by-trial choices and RTs. From this, and after getting the best-fitting model parameters, we can get the model prediction of choice probability (**Equation 9**) and mean RT (**Equation 10**) using the following integrals, respectively:

$$p\left(\text{choice} = i\right) = \int_0^{+\infty} L\left(i, t\right) dt \tag{9}$$

$$\mathbf{E}\left(t_i\right) = \int_0^{+\infty} \left(tL\left(i, t\right)\right) dt \Big/ \int_0^{+\infty} L\left(i, t\right) dt \tag{10}$$

In practice, we used the trapezoidal numerical integration with a small enough $dt = 0.001$ s and a large enough $T_{max} = 100$ s. Empirically, no negative drift rates were observed in any of the model parameter estimates in any participant.

## Context-independent model

The utility function $U_i$ in the context-independent model is 'blind' to the distractor (D). For ternary trials, the context-independent model only takes H and L as inputs. For instance, the context-independent AU model assumes $AU_H = (\lambda)HX + (1 − \lambda)HP$ and $AU_L = (\lambda)LX + (1 − \lambda)LP$, with $\lambda$ being the attribute weight of reward magnitude ($0 \leq \lambda \leq 1$).

## SI model

We adopted the simplest form of SI model. The SI model discards information about those choice alternatives with relatively lower value in each attribute separately (*Usher et al., 2019*). Alternatives are compared and then selectively gated based on their ranks within each attribute. The highest attribute value remains unchanged ($w_1 = 0$), intermediate value gets suppressed by a multiplicative weight $w_2$, whilst the lowest value gets more strongly suppressed by $w_3$ ($0 \leq w_2 \leq w_3 \leq 1$). That is, the SI $w$ represents the percent reduction in information processing gain. In case of a tie for the highest value, the two high values are unchanged, and the lowest value is suppressed by $w_3$. By contrast, in case of a tie for the lowest value, the two low values both get suppressed by $w_3$. After the selective gating within each attribute, which transforms attribute values to $X'$ and $P'$, respectively, an AU is constructed for each alternative via a weighted averaging, that is, $AU = (\lambda)X' + (1 − \lambda)P'$, with $\lambda$ being the attribute weight of reward magnitude ($0 \leq \lambda \leq 1$). This AU (after SI) is then fed into the FFI model as the component U for each alternative (*Equation 3*).

## Adaptive gain model

The attribute value of a certain alternative, $i$, for example, the reward magnitude, $X_i$, is inhibited by a context mean across all alternatives in a choice-set (e.g., $n$ alternatives), which yields $\bar{X}_i$. $\bar{X}_i$ then passes through a sigmoidal transduction function with slope $s$ and bias term $b$ (*Dumbalska et al., 2020*). This gain control is adaptive in that it accommodates the contextual information. Importantly, the gain control operates independently on each attribute, and it produces the final subjective attribute value ($X_i^{AG}$) for the construction of AU for each alternative:

$$X_i^{AG} = 1 \Big/ \left(1 + e^{\frac{b_x − \bar{X}_i}{s_x}}\right) \tag{11}$$

$$\bar{X}_i = X_i − \left(\frac{\sum_i X_i}{n}\right) \tag{12}$$

The adaptive gain modulation of $P$ can be obtained by replacing $X$ with $P$ in the above two equations. Ultimately, the utility of each alternative is computed as a weighted average across attribute values:

$$U_i = \lambda X_i^{AG} + \left(1 − \lambda\right) P_i^{AG} \tag{13}$$

## Dual-route model

The dual-route model arbitrates between a vanilla MI process and an MI process with additional DN (*Chau et al., 2020*). In the vanilla MI 'route', this model assumes that the evidence input to the accumulator is:

$$I_i = U_i - f_{MI}\left(\frac{\sum_i U_i}{n}\right) \tag{14}$$

where $U$ is expected value (EV), that is, EV = XP, and $f_{MI}$ is the inhibition strength of MI. In another rival route, the utility of each choice alternative is additionally divisively normalised by the sum of utilities across all alternatives (H + L in a binary trial; H + L + D in a ternary trial):

$$I_i^{DN} = U_i^{DN} - f_{MI}\left(\frac{\sum_i U_i^{DN}}{n}\right) \tag{15}$$

$$U_i^{DN} = U_i / \sum_i U_i \tag{16}$$

The drift rate takes the same form as that in FFI: $\mu_i = kI_i + I_0$. Based on *Equation 8* and for clarity, we rewrite the two components in *Equation 8* as **g** and **G**:

$$\mathbf{G}\left(\mu, T, \theta, \sigma\right) = 1 - \Phi\left(\frac{\mu T - \theta}{\sigma\sqrt{T}}\right) + e^{\frac{2\theta\mu}{\sigma^2}}\Phi\left(\frac{-\mu T - \theta}{\sigma\sqrt{T}}\right) \tag{17}$$

$$\mathbf{g}\left(\mu, T, \theta, \sigma\right) = \frac{\theta}{\sigma\sqrt{2\pi T^3}}e^{\frac{-(\theta - \mu T)^2}{2\sigma^2 T}} \tag{18}$$

The likelihood of the DN route and the vanilla route arriving at the decision bound first to yield a choice $i$ is, respectively:

$$L^{DN}\left(i, T\right) = \mathbf{g}\left(\mu_i^{DN}, T, \theta, \sigma\right)\mathbf{G}\left(\mu_j^{DN}, T, \theta, \sigma\right)\mathbf{G}\left(\mu_i, T, \theta, \sigma\right)\mathbf{G}\left(\mu_j, T, \theta, \sigma\right) \tag{19}$$

$$L\left(i, T\right) = \mathbf{g}\left(\mu_i, T, \theta, \sigma\right)\mathbf{G}\left(\mu_j, T, \theta, \sigma\right)\mathbf{G}\left(\mu_i^{DN}, T, \theta, \sigma\right)\mathbf{G}\left(\mu_j^{DN}, T, \theta, \sigma\right) \tag{20}$$

The final likelihood function of choice $i$ at time $T$ thus is: $L^{DN}\left(i, T\right) + L\left(i, T\right)$. These closed-form analytical solutions of choice probability and mean RT, as given by *Equation 9*, *Equation 10* match well with the model simulations performed in *Chau et al., 2020*. Empirically, the trial-by-trial model predictions of a model with free $\sigma$ (one for each route) vs. a model with $\sigma = 1$ are highly similar (across-participants mean Pearson's correlation coefficient $r(148) = 0.96$ for ternary-choice probabilities, $r(148) = 0.98$ for binary-choice probabilities, $r(148) = 0.97$ for ternary RTs, $r(148) = 0.88$ for binary RTs). Here, we chose the simpler model with unit $\sigma$ (lower BIC compared to the free $\sigma$ model: −169 for ternary trials and −790 for binary trials, summed across participants).

### Static models
The static models were fit to trial-by-trial human choice probability (while ignoring RT). These models assume a softmax function that takes as inputs the options' utilities (AU, EV, or divisively normalised EV, see *Figure 3*) and controls the level of decision noise via an inverse temperature parameter $\beta$:

$$p\left(\text{choice} = i\right) = \frac{e^{\beta U_i}}{\sum_i e^{\beta U_i}} \tag{21}$$

The model-predicted and the empirical choice probabilities were then used to calculate the binomial log-likelihood: $LL \propto p_e \log\left(p_m\right) + \left(1 - p_e\right)\log\left(1 - p_m\right)$, in which $p_m$ and $p_e$ denotes the model-predicted and empirical $p$(H over L), respectively.

### Model fitting, comparison, and recovery
Models were fit to data via maximising the log-likelihood summed over trials. Because the *LL* functions all have analytical solutions, Matlab's fmincon was used for parameter optimisation. To avoid local optima, we refit each model to each participant's behavioural data at least 10 times using a grid of randomly generated starting values for the free parameters. Conversative criteria were used during the optimisation search: MaxFunEvals = MaxIter = 5000; TolFun = TolX = $10^{-10}$. We then calculated each model's posterior frequency and protected exceedance probability (i.e., the probability corrected for chance level that a model is more likely than any others in describing the data) using

the variational Bayesian analysis (VBA) toolbox (*Rigoux et al., 2014*; *Daunizeau et al., 2014*). To be consistent with the methods in *Chau et al., 2020*, we fit models to those trials with either H or L responses whilst ignoring a very small portion of trials in which D was accidentally chosen (median proportion of D choices: 4%, IQR: [2%, 8%], across participants).

Model comparison relied on cross-validation. That is, for each participant, we split the 150 trials into five folds and then fit each model to a 'training' set comprising four random folds of trials, obtained the best-fitting parameters, and finally calculated the *LL* summed across trials in the left-out 'test' fold. This process was repeated over test folds and the final cross-validated *LL* was computed as the mean *LL* across cross-validation folds. Model recovery was carried out by first simulating each model's behaviour (choice probability and RT) with its own best-fitting parameters obtained from fitting the model to each participant's raw data; then cross-fitting all models of interest to the simulated behaviour and calculating the *LL*s summed over trials; and finally comparing models' goodness-of-fit using Bayesian model comparison.

## Subjective distortion of reward attributes

When modelling binary-choice behaviour, in addition to assuming linear subjective representations of reward attributes for simplicity, we also considered non-linear subjective functions (*Zhang and Maloney, 2012*). The probability distortion undergoes a log-odds function (*Equation 22*) that converts a probability value $P$ to $P_d$, in a flexible manner (S-shape, inverted S-shape, concave, or convex, depending on parameterisation), whereas the magnitude distortion follows a power-law function (*Equation 23*) that converts $X$ to $X_d$:

$$\log\left(\frac{P_d}{1-P_d}\right) = \eta \log\left(\frac{P}{1-P}\right) + (1-\eta)\log\left(\frac{P_0}{1-P_0}\right) \tag{22}$$

$$X_d = X^\gamma \tag{23}$$

where $\eta > 0$, $P_0$ ($0 < P_0 < 1$), and $\gamma > 0$ are the three parameters controlling the extent of non-linearity of the distortion functions.

In addition to the log-odds function, we also considered another non-linear function (only 1 free parameter $\tau$) for the probability distortion, described by *Equation 24*:

$$P_d = \frac{P^\tau}{\left[P^\tau + (1-P)^\tau\right]^{\frac{1}{\tau}}} \tag{24}$$

This particular function underlies the subjective probability distortion in a model of the *prospect theory* (*Kahneman and Tversky, 1979*). More specifically, the prospect-theory model assumes a multiplicative utility that multiplies $X_d$ (*Equation 23*) with $P_d$ (*Equation 24*). Another popular multiplicative model is the standard *expected utility* (*Von Neumann and Morgenstern, 2007*) in which the probability function is linear (i.e., $\tau$ is fixed to 1) while $\gamma$ governing $X_d$ is a free parameter. We included these 2 additional multiplicative models when comparing additive vs. multiplicative strategies (*Figure 3—figure supplement 1*).

## Condition-unspecific response bias correction

Choice behaviour can differ between a binary condition and a ternary condition, simply due to the increased number of options (higher cognitive demands) regardless what they are. To estimate these condition-unspecific biases, we used a permutation approach to shuffling the mappings between the ternary conditions and their matched binary conditions (5000 permutations). This yielded a mean bias term for the 'T minus matched B' relative accuracy data, and was eventually removed from the re-referenced ternary-trial performance ('T – B').

## GLM analysis of relative choice accuracy

We used GLM approach to analysing choice behaviour. For ternary conditions, we focused on the *relative* accuracy, that is, the proportion of H choices among trials with H or L choices (while ignoring a small number of D-choice trials), which has been commonly used as a measure of the violation of the IIA principle. Matched binary trials were identified for each ternary trial (see *Figure 2—figure supplement 1*). Among the 150 ternary trials there are 149 unique ternary conditions with a unique

combination of probability (*P*) and magnitude (*X*) attributes across the three alternatives (H, L, and D), whereas among the 150 binary trials there are only 95 unique conditions with a specific combination of *P* and *X* across the two available options (H and L). The ternary and binary trials do not have 'one-to-one' mapping. Some ternary trials have more than 1 matched binary trial. The different counts of matched binary trials were explicitly considered as 'observation weights' in the GLM. We used logit binomial GLM to analyse trial-by-trial relative choice accuracy (Matlab's glmfit with 'binomial' distribution and 'logit' link function). The GLMs in *Figure 2* (d, e; using relative distractor variable 'DV – HV') and *Figure 2—figure supplement 2* (using absolute distractor variable 'DV') were specified as follows:

$$\text{Logit}(p(\text{H over L})) \sim \beta_0 + \beta_1\, z(\text{HV} - \text{LV}) + \beta_2\, z(\text{DV} - \text{HV}) + \beta_3\, z(\text{HV} - \text{LV}) \times z(\text{DV} - \text{HV})$$

$$\text{Logit}(p(\text{H over L})) \sim \beta_0 + \beta_1\, z(\text{HV} - \text{LV}) + \beta_2\, z(DV) + \beta_3\, z(\text{HV} - \text{LV}) \times z(DV)$$

$z(\cdot)$: *z*-score normalisation of the regressor vector. The interaction terms were constructed *after* individual component terms were *z*-scored (*Gluth et al., 2018*). The analysis shown in *Figure 2f* combined ternary (T) and binary (B) relative accuracies in a single GLM and assessed how distractor-related effects interacted with 'Condition' (i.e., 'C', a dummy variable: 0 for B, 1 for T). The code for reproducing *Figure 2* and *Figure 2—figure supplements 2 and 3*, GLMs can be found here: https://github.com/YinanCao/multiattribute-distractor, (*Cao, 2022* copy archived at swh:1:rev:a465f0c394fa1b29b16ff8aa7d384f38f0a0c67b).

## Construction of binned maps

The choice-accuracy and RT maps (*Figure 2b, c*; *Figure 3a*) were constructed using the exact binning approach of *Chau et al., 2020*. Data were averaged using a sliding square window inside the space defined by the two expected-value differences, (DV – HV) and (HV – LV). The edge length of the window is 30% quantile difference along each dimension, with step-size being equal to 1% quantile difference. This binning approach smooths the data and results in a map of size 71-by-71.

## Acknowledgements

We thank the authors of the original studies for sharing their data publicly (*Chau et al., 2014*: https://doi.org/10.5061/dryad.040h9t7; *Gluth et al., 2018*: https://osf.io/8r4fh/). We thank Rani Moran and Marius Usher for helpful discussions on an earlier version of this paper. This work was supported by the EU Horizon 2020 Research and Innovation Program (ERC starting grant no. 802905) to KT. The funders had no role in study design, data collection, and analysis, decision to publish or preparation of the manuscript.

## Additional information

### Funding

| Funder | Grant reference number | Author |
|---|---|---|
| European Research Council | EU Horizon 2020 Research and Innovation Program (ERC starting grant no. 802905) | Konstantinos Tsetsos |

The funders had no role in study design, data collection, and interpretation, or the decision to submit the work for publication.

### Author contributions

Yinan Cao, Conceptualization, Resources, Data curation, Software, Formal analysis, Validation, Visualization, Methodology, Writing – original draft, Writing – review and editing; Konstantinos Tsetsos, Conceptualization, Resources, Supervision, Funding acquisition, Writing – original draft, Writing – review and editing

### Author ORCIDs

Yinan Cao ⬥ http://orcid.org/0000-0002-9881-5106

Konstantinos Tsetsos ⬤ http://orcid.org/0000-0003-2709-7634

### Ethics

The current manuscript re-analyses previously published datasets, thus no data have been generated for this manuscript. The relevant information about ethical approvals of these published datasets can be found in the original studies.

### Decision letter and Author response

Decision letter https://doi.org/10.7554/eLife.83316.sa1
Author response https://doi.org/10.7554/eLife.83316.sa2

## Additional files

### Supplementary files

- Supplementary file 1. Optimal parameter estimates of dynamic models: mean (SE) and cross-validated log-likelihood (CV LL).
- Supplementary file 2. Optimal parameter estimates of static models: mean (SE) and cross-validated log-likelihood (CV LL).
- Supplementary file 3. Generalised linear models (GLMs) using subjective additive utility (AU) regressors. *Significant effects (p < 0.05; two-sided one-sample $t$-tests of GLM coefficients against 0) following Holm's sequential Bonferroni correction for multiple comparisons. C: 'Condition' (binary vs. ternary).
- Supplementary file 4. Optimal parameter estimates of context-dependent dynamic models: mean (SE) and cross-validated log-likelihood (CV LL).
- MDAR checklist

### Data availability

The current manuscript re-analyses previously published datasets, so no new data have been generated for this manuscript. Analysis/computational modelling code has been uploaded to GitHub: https://github.com/YinanCao/multiattribute-distractor (copy archived at swh:1:rev:a465f0c394fa1b29b16ff8aa7d384f38f0a0c67b).

The following previously published datasets were used:

| Author(s) | Year | Dataset title | Dataset URL | Database and Identifier |
|---|---|---|---|---|
| Chau B, Kolling N, Hunt L, Walton M, Rushworth M | 2020 | Data from: A neural mechanism underlying failure of optimal choice with multiple alternatives | https://doi.org/10.5061/dryad.040h9t7 | Dryad Digital Repository, 10.5061/dryad.040h9t7 |
| Gluth S, Spektor M, Rieskamp J | 2018 | Data from: Value-based attentional capture affects multi-alternative decision making | https://osf.io/8r4fh/ | Open Science Framework, 8r4fh |

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
