## [Editor Report]

This study presents an important finding on the decoy effect in multiattribute economic choices in humans. It makes a compelling case for the conclusion that the distractor effect reported in previous articles was confounded with the additive utility difference between the available alternatives. Though the contribution is somewhat narrowly focused with respect to the phenomenon that it addresses – the distractor effect in risky choice – it is important for understanding this particular phenomenon.

---

## [Decision Letter]

**Decision letter after peer review:**

Thank you for submitting your article "Clarifying the role of an unavailable distractor in human multiattribute choice" for consideration by *eLife*. Your article has been reviewed by 3 peer reviewers, and the evaluation has been overseen by a Reviewing Editor and Joshua Gold as the Senior Editor. The following individual involved in review of your submission has agreed to reveal their identity: Jerome Busemeyer (Reviewer #2).

Essential revisions:

1) We understand this is a complex topic, but it seems there is still much space for the clarity of the paper to be improved. Some focusing and separation of main issues from minor details may help. Both Reviewers #1 and #2 provide good suggestions on writing.

2) Which computation (addition or multiplication) humans perform for value and probability in decision under risk is an essential claim of the paper. The "multiplicative" models such as the prospect theory models (with varying utility and probability weighting functions) have proved to be the best model for decision under risk in many studies. Even though the "additive" model can explain some qualitative features of the data that would be hardly explainable for "multiplicative" models in this particular situation, it would be premature to reject "multiplicative" models without including the most representative of them (i.e., prospect theory models) in the model comparison. The authors are encouraged to consider such models (in addition to the EV model) or to explain why not including such models in their analysis. Please see the comments of Reviewers #1 and #2.

3) The authors are also encouraged to have a more in-depth discussion of the limitation of using the additive rather than the multiplicative rule for decision under risk. Please see the comment of Reviewer #2 for details.

4) To include some discussion of the neurobiological basis of the work would be helpful for the potential readers. Please see the comment of Reviewer #3 for details.

*Reviewer #1 (Recommendations for the authors):*

In this study, the authors investigate multi-attribute economic decisions in humans. More specifically, they investigate the impact of adding a decoy option to a binary choice, the effect of which has been debated in the previous literature with recent studies finding effects in different directions (negative, positive, none, both). By re-analyzing large datasets from some of these studies (Chau et al. Nat Neuro 2014; Gluth et al. *eLife* 2018; Chau et al. *eLife* 2020) and by applying computational modeling to the data, the authors propose that a subjective encoding rule comparing attributes separately, and not computed as one multiplicative value, explains participants' behavior on different levels of granularity. Context effects were well captured by a selective integration model. The work has many positive features, including praising open science, the analysis of the potential confounds of previously used designs, and both quantitative and qualitative comparisons of several well-justified models. Overall, the paper is well-written and the findings add a new perspective to the field. I have however some comments and suggestions for the authors, but in general I am very positive about their interesting work.

1) The term "additive utility" might be confusing, since risk tendencies are not addressed in the models. Would the multiplicative rule be named "multiplicative utility" instead of expected value? In any case, AU is a central point of the paper and the term needs to be defined in a more explicit way, much sooner in the paper (AU is mentioned on line 121 but is only defined in the next paragraph). It is then sometimes unclear, when the "value" is mentioned, which form of computed value it is (e.g., line 123).

2) Related to my previous point, is there any reason why the authors chose not to include such models in their analysis (e.g., "standard" utility, prospect theory) in addition to the EV model?

3) The analysis of the binary vs. trinary trials, highlighting a design confound which leads to spurious effects, was very good in my opinion. However, this part is a bit confusing in its current form, leaving the reader to wonder how such a result can appear (finding a distractor effect in binary trials, where there is no distractor; e.g., line 138, line 238). The manuscript will benefit from a better definition of what the authors mean by a distractor effect in binary trials.

4) It is not clear whether the dataset from Chau et al. 2020 was included in the analysis. The methods section does not include this dataset but the acknowledgement section seems to suggest otherwise. If this dataset wasn't included, is there any reason why the authors chose to leave it out?

*Reviewer #2 (Recommendations for the authors):*

Figure 1

What black dot? I see a dark green one.

Figure f is not clear’

Fig 2

These figures need more description, I don’t quite understand what is H1 and H2 in h,i.?

What is the diff between h and i in this figure?

pages 407-423

I don't have a clear idea about this bias correction.

Equation 4

What is the stopping rule?

Are these independent accumulators racing to a bound ?

Equation 4 is based on this assumption.

*Reviewer #3 (Recommendations for the authors):*

Considering the readership of *eLife*, I would recommend a paragraph to discuss the neurobiological basis of your work, for example, how your work may relate to the following questions: how multiple features are represented in the brain (e.g., Rigotti, Nature, 2013; Flesch et al., Neuron, 2022; Fusi et al., Curr. Opin. Neurobiol., 2016), and the neural mechanism of multi-alternative decision making (e.g. Albantakis and Deco, PNAS, 2009; Churchland and Ditterich, Curr. Opin. Neurobiol., 2012).

It was not proper to put "not multiplicative" in the second title of Result (Line 242) since you did not provide any evidence against "multiplicative" integration in the immediate following paragraph.

---

## [Author Response]

Essential revisions:1) We understand this is a complex topic, but it seems there is still much space for the clarity of the paper to be improved. Some focusing and separation of main issues from minor details may help. Both Reviewers #1 and #2 provide good suggestions on writing.

Thank you for this advice. We have now followed the writing suggestions from the reviewers to clarify the definitions of the key terms in this paper. We have also updated Figure 1 and the relevant captions to better illustrate the key issues that this paper addresses.

2) Which computation (addition or multiplication) humans perform for value and probability in decision under risk is an essential claim of the paper. The "multiplicative" models such as the prospect theory models (with varying utility and probability weighting functions) have proved to be the best model for decision under risk in many studies. Even though the "additive" model can explain some qualitative features of the data that would be hardly explainable for "multiplicative" models in this particular situation, it would be premature to reject "multiplicative" models without including the most representative of them (i.e., prospect theory models) in the model comparison. The authors are encouraged to consider such models (in addition to the EV model) or to explain why not including such models in their analysis. Please see the comments of Reviewers #1 and #2.

Thank you for suggesting to examine in more depth additional multiplicative models. In response to the above point (2) of the *Essential Revision* and a specific comment (2) from Reviewer #1, we have now added the additional computational models as requested: the *prospect-theory* model (Kahneman and Tversky, 1979) and the standard *expected-utility* model (Von Neumann and Morgenstern, 1953). Unlike the ‘vanilla’ expected value (EV) model, these two additional *multiplicative* models allow for non-linear functions of reward magnitude (X) and probability (P). We describe the reward and probability functions relevant for these two multiplicative models as: Xd=XγPd=Pτ[Pτ+(1−P)τ]1τ, where equation 1 describes the reward magnitude (X) function (i.e., the *utility function*) and equation 2 specifies the probability (P) function (subscript *d*: ‘distortion’); γ>0 and τ>0 are 2 free parameters controlling the extent of non-linearity of the X and P functions, respectively. In the prospect-theory model, both γ and τ are free parameters, whereas in the expected-utility model (assuming the classic von Neumann-Morgenstern power-law utility function) only γ is a free parameter whilst τ=1 (i.e., the expected-utility model assumes a linear probability function without distortion).

We employed a standard Bayesian random-effects analysis, using the models’ cross-validated log-likelihood to compute the protected exceedance probability (*P*_pexc_) that a given cross-validated model fits participants’ data better than other models. We found that, indeed, the prospect-theory model and the expected-utility model fit the binary-choice data better than the vanilla EV model did (Figure 3—figure supplement 1). However, we found that the AU model (without any non-linear distortion of reward attributes) fit the data decisively better than all multiplicative models (*P*_pexc_ > 0.97) in both the dynamic and the static versions of the models [cross-validated log-likelihood ratio > 18.26, ΔBIC < -95 (AU minus others) in the dynamic version; cross-validated log-likelihood ratio > 56.42, ΔBIC < -145 in the static version; metrics were pooled over participants]. We have also briefly mentioned these new results in the Results section, and described the technical details in the Methods section of the revised manuscript. Code for replicating these results: https://github.com/YinanCao/multiattribute-distractor/

3) The authors are also encouraged to have a more in-depth discussion of the limitation of using the additive rather than the multiplicative rule for decision under risk. Please see the comment of Reviewer #2 for details.

Thanks for raising this important point. We agree with the editors and Reviewer #2 that it is important to discuss in more depth the pros and cons of additive vs. multiplicative rules in risky decision-making. We have now extensively expanded our discussion on the integration rule in the fourth paragraph of our Discussion section (starting with ‘Notably, and contrary to the normative rule…’). We hope the added discussion points convey the following messages: (1) we do not claim that the additive rule can or should be a domain-general principle that generalises across all types of risky choice task, and (2) yet, researchers should be aware of this strategy as a viable alternative to the multiplicative strategy that humans (Rouault et al., 2019) and other animals (Farashahi et al., 2019; Bongioanni et al., 2021) might be adopting when solving certain risky choice problems.

4) To include some discussion of the neurobiological basis of the work would be helpful for the potential readers. Please see the comment of Reviewer #3 for details.

We have now discussed the relevant neural basis of the decision strategies in our work. Specifically, we summarised evidence in past reports pertaining to the neural coding of multiplicative vs. additive integration of reward attributes in the second last paragraph of our Discussion.

Reviewer #1 (Recommendations for the authors):In this study, the authors investigate multi-attribute economic decisions in humans. More specifically, they investigate the impact of adding a decoy option to a binary choice, the effect of which has been debated in the previous literature with recent studies finding effects in different directions (negative, positive, none, both). By re-analyzing large datasets from some of these studies (Chau et al. Nat Neuro 2014; Gluth et al. eLife 2018; Chau et al. eLife 2020) and by applying computational modeling to the data, the authors propose that a subjective encoding rule comparing attributes separately, and not computed as one multiplicative value, explains participants' behavior on different levels of granularity. Context effects were well captured by a selective integration model. The work has many positive features, including praising open science, the analysis of the potential confounds of previously used designs, and both quantitative and qualitative comparisons of several well-justified models. Overall, the paper is well-written and the findings add a new perspective to the field. I have however some comments and suggestions for the authors, but in general I am very positive about their interesting work.1) The term "additive utility" might be confusing, since risk tendencies are not addressed in the models. Would the multiplicative rule be named "multiplicative utility" instead of expected value? In any case, AU is a central point of the paper and the term needs to be defined in a more explicit way, much sooner in the paper (AU is mentioned on line 121 but is only defined in the next paragraph). It is then sometimes unclear, when the "value" is mentioned, which form of computed value it is (e.g., line 123).

We have now better defined these terms in our revised manuscript. In general, we used the term ‘utility’ to refer to the *subjective attractiveness* of a prospect, which is the outcome that can be used for guiding a choice after combining multiple reward-related attributes into a net quantity. We have now explicitly defined ‘additive utility’ early on in our Introduction (lines 123 ~ 125, see excerpt below) and ensured this definition is repeated in the Results section. Meanwhile, we also explicitly use ‘expected value’ (EV) instead of a fuzzy term ‘value’ in our Introduction and Results when describing ‘reward magnitude x probability’ (with no non-linear transformation) of a risky prospect.

Lines 123 ~ 125 (Introduction):

“For instance, if people valuate prospects by simply adding up their payoff and probability information (hereafter *additive utility* or *AU*), …”

2) Related to my previous point, is there any reason why the authors chose not to include such models in their analysis (e.g., "standard" utility, prospect theory) in addition to the EV model?

Thanks for this comment. We have now included two additional multiplicative models as requested (see detailed analyses above). The Bayesian model comparison showed that the AU model prevailed over multiplicative models (standard expected-utility model, prospect-theory model, and EV). We briefly discuss this model comparison in the Results section, linking to the new Figure 3—figure supplement 1 that summarises this new result.

3) The analysis of the binary vs. trinary trials, highlighting a design confound which leads to spurious effects, was very good in my opinion. However, this part is a bit confusing in its current form, leaving the reader to wonder how such a result can appear (finding a distractor effect in binary trials, where there is no distractor; e.g., line 138, line 238). The manuscript will benefit from a better definition of what the authors mean by a distractor effect in binary trials.

We have now used the more explicit term ‘notional distractor effect’ when referring to this paradoxical effect in binary choice. We use ‘notional’ to make it clear the idea that there should be strictly no effect of the distractor because the distractor did not exist, thus is only *notional*, in binary trials. We believe this clarification will help the readers better distinguish this paradoxical effect in binary trials from the distractor effect in ternary trials. Correspondingly, we also updated Figure 2e and g, in which the ‘notional distractor’ effects are marked. Please see a few text excerpts below in our revised manuscript:

Lines: 197 ~ 198 (Results):

“Given that participants never saw D in binary trials, the distractor variable is notional and should have no effect on the binary baseline accuracies.”

Lines 201 ~ 204 (Results):

“We dub any effect that D has upon binary-choice accuracy as the ‘notional distractor effect’. We emphasise here that a notional distractor effect is not a genuine empirical phenomenon but a tool to diagnose target-/distractor-related covariations in the experimental design.”

Lines 251 ~ 253 (Results):

“These results equate the distractor effects in ternary trials (D present) with the notional distractor effects in binary trials (D absent), indicating that the former arose for reasons other than the properties of D.”

4) It is not clear whether the dataset from Chau et al. 2020 was included in the analysis. The methods section does not include this dataset but the acknowledgement section seems to suggest otherwise. If this dataset wasn't included, is there any reason why the authors chose to leave it out?

The reviewer was right that the study by Chau et al. (2020; *eLife*) collected a new dataset (‘Hong-Kong’) of this speeded-risky-choice task (*N* = 40). However, it did not include binary-choice trials in the experimental design, and thus unfortunately could not be analysed and combined with other datasets in the same fashion. To avoid confusion, we have now removed the citation of Chau et al. 2020 from the Acknowledgements in our revised manuscript, and also explained this point in the first paragraph of the Methods section.

Reviewer #2 (Recommendations for the authors):Figure 1What black dot? I see a dark green one.Figure f is not clear’

We have thoroughly revised Figure 1. This figure now begins with illustrating context effect in perception (Figure 1a), and then illustrating two opposing unidimensional distractor effects in value-based choice reported in past research (Figure 1b), and finally using two examples to illustrate target- and distractor-related covariations as potential confounds for distractor effects (panels c and d).

Figure 2These figures need more description, I don’t quite understand what is H1 and H2 in h,i.?What is the diff between h and i in this figure?

Thanks for the comments. We have now changed these two figures and updated the captions to make things clearer.

pages 407-423I don't have a clear idea about this bias correction.

Thanks for this feedback. We now further clarified our method when describing this analysis in the Results: “We estimated this ‘generic’ bias by a permutation procedure whereby we randomly shuffled the true mappings between ternary conditions and their matched binary conditions and obtained an average T – B accuracy contrast across random permutations (Figure 4c; see Methods).”

Equation 4What is the stopping rule?Are these independent accumulators racing to a bound ?Equation 4 is based on this assumption.

These are indeed accumulators racing to a bound with the ‘drift’ of each accumulator being flexibly defined (see equation 3 in our Methods section; for feedforward-inhibition strength c = 0 accumulators are independent; for c = 1 the model behaves akin to a diffusion model).

Reviewer #3 (Recommendations for the authors):Considering the readership of eLife, I would recommend a paragraph to discuss the neurobiological basis of your work, for example, how your work may relate to the following questions: how multiple features are represented in the brain (e.g., Rigotti, Nature, 2013; Flesch et al., Neuron, 2022; Fusi et al., Curr. Opin. Neurobiol., 2016), and the neural mechanism of multi-alternative decision making (e.g. Albantakis and Deco, PNAS, 2009; Churchland and Ditterich, Curr. Opin. Neurobiol., 2012).

Thank you for this comment. We have now added a dedicated paragraph in our Discussion (2nd last paragraph) where we talk about the relevant neural underpinnings of the decision strategies and feature (attribute) representations involved in our work. Upfront, we did not extend our discussion to the neural mechanisms of multialternative decision-making. We found it challenging to do so in an authoritative fashion, since the key task we focused on here involves two available alternatives and one unavailable alternative (i.e., it is technically a binary choice).

It was not proper to put "not multiplicative" in the second title of Result (Line 242) since you did not provide any evidence against "multiplicative" integration in the immediate following paragraph.

Thanks for this comment. We have now revised the subtitle of that section in the Results from ‘Additive and not multiplicative information integration in risky choice’ to ‘Integrating reward and probability information additively in risky choice’.

References

Bongioanni, A. et al. Activation and disruption of a neural mechanism for novel choice in monkeys. *Nature* 591, 270–274 (2021).

Farashahi, S., Donahue, C. H., Hayden, B. Y., Lee, D. and Soltani, A. Flexible combination of reward information across primates. *Nat Hum Behav* 3, 1215–1224 (2019).

Kahneman, D. and Tversky, A. Prospect Theory: An analysis of decisions under risk. *Econometrica* 47, 278 (1979).

Rouault, M., Drugowitsch, J. and Koechlin, E. Prefrontal mechanisms combining rewards and beliefs in human decision-making. *Nat. Commun*. 10, 301 (2019).

von Neumann, J. and Morgenstern, O. *Theory of Games and Economic Behavior*. (Princeton University Press, 1953).